# SOME ROBUSTNESS PROPERTIES OF LABEL CLEANING

## ABSTRACT

We demonstrate that learning procedures that rely on aggregated labels, e.g., label information distilled from noisy responses, enjoy robustness properties impossible without data cleaning. This robustness appears in several ways. In the context of risk consistency—when one takes the standard approach in machine learning of minimizing a surrogate (typically convex) loss in place of a desired task loss (such as the zero-one mis-classification error)—procedures using label aggregation obtain stronger consistency guarantees than those even possible using raw labels. And while classical statistical scenarios of fitting perfectly-specified models suggest that incorporating all possible information—modeling uncertainty in labels—is statistically efficient, consistency fails for "standard" approaches as soon as a loss to be minimized is even slightly mis-specified. Yet procedures leveraging aggregated information still converge to optimal classifiers, highlighting how incorporating a fuller view of the data analysis pipeline, from collection to model-fitting to prediction time, can yield a more robust methodology by refining noisy signals.

## 1 INTRODUCTION

Consider the data collection pipeline in a supervised learning problem. Naively, we say that we collect pairs $(X_i, Y_i)_{i=1}^n$ of features $X_i$ and labels $Y_i$, fit a model, and away we go Hastie et al. (2009). But this belies the complexity of modern datasets Deng et al. (2009); Krizhevsky and Hinton (2009); Russakovsky et al. (2015), which require substantial data cleaning, filtering, often crowdsourcing multiple labels and then denoising them. The crowdsourcing community has intensively studied such data cleaning, especially in the context of obtaining "gold standard" labels Dawid and Skene (1979); Whitehill et al. (2009); Welinder et al. (2010); Vaughan (2018); Platanios et al. (2020). We take a complementary view of this process, investigating the ways in which data aggregation fundamentally and necessarily improves the consistency of models we fit.

In a sense, this paper argues that label cleaning, or aggregating labels together, provides robustness that is impossible to achieve without aggregating labels. There are two faces to this robustness. First, we improve consistency of estimation: when minimizing a surrogate loss (e.g., the multiclass logistic loss) instead of a *task* loss (e.g., the zero-one error), procedures that use aggregated labels can achieve consistent and optimal prediction in the limit when this is impossible without data aggregation. Second, even in finite-dimensional statistical problems, this aggregation can provide consistent classifiers when standard methods fail.

Important contributions to the theory of surrogate risk consistency trace to the 2000s Zhang (2004c); Lugosi and Vayatis (2004); Steinwart (2007), with Bartlett, Jordan, and McAuliffe (2006) characterizing when fitting a model using a convex surrogate is consistent for binary classification for the zero-one error. Since this work, there has been an abundance of work on surrogate risk consistency, including on multi-label classification Zhang (2004a); Tewari and Bartlett (2007); Gao and Zhou (2011); Zhang and Agarwal (2020); Awasthi et al. (2021), ranking problems Duchi et al. (2010; 2012); Pires et al. (2013), structured prediction Osokin et al. (2017); Cabannes et al. (2020); Nowak-Vila et al. (2020), ordinal regression Pedregosa et al. (2017), and general theory Steinwart (2007). On the one hand, these analyses, which consider the standard supervised learning scenario of data pairs $(X, Y)$, enable us to fully exploit the entire statistical theory of empirical processes (van der Vaart and Wellner, 1996; Bartlett et al., 2005; Koltchinskii, 2006; Bartlett et al., 2006). On the other, they do not address the data aggregation machinery now common in modern dataset creation.

It is thus natural to ask about the interaction between consistency and data aggregation—to begin with, do we need to aggregate at all? If we can achieve surrogate consistency without data aggregation,

we should perhaps just rely on our mature theoretical understanding of processes with $(X, Y)$ pairs. Going one step further, if we aggregate, does aggregation help consistency, and in what sense does it help? These main questions motivate this paper.

To further underpin the importance of studying consistency and aggregated labels, we propose concrete examples—in ranking and binary- and multiclass-classification with linear estimators— where estimators using only pairs $(X, Y)$ necessarily fail, but label aggregation methods yield consistency. We develop new notions and theory for surrogate consistency with data aggregation. In fully nonparametric scenarios, we show how the number of samples aggregated combine with noise conditions to improve consistency. Aggregation will also allow us to demonstrate surrogate risk consistency under only weak conditions the surrogate loss; in the language of the field, losses using aggregated labels admit (approximate) linear comparison inequalities. Additionally, in contrast to conventional risk consistency theory, which requires taking a hypothesis class $\mathcal{F}$ consisting of all measurable functions, we will show results in classification problems where aggregating labels guarantees consistency even over restricted hypothesis classes, which may fail without aggregation.

## 2 Preliminaries

We first review classical surrogate risk minimization. Let $\mathcal{X}$ be the input space and $\mathcal{Y}$ be the output space, with data $(X_i, Y_i)_{i=1}^n \in \mathcal{X} \times \mathcal{Y}$ drawn i.i.d. $P$. Consider learning a scoring function $f : \mathcal{X} \to \mathbb{R}^d$ that maps an input $x \in \mathcal{X}$ to a score $s \in \mathbb{R}^d$ for some $d \geq 1$, where a decoder $\mathsf{d} : \mathbb{R}^d \to \mathcal{Y}$ determines the final prediction via $\widehat{y} = \mathsf{d} \circ f(x)$. Given a loss $\ell : \mathcal{Y} \times \mathcal{Y} \to \mathbb{R}_+$ and hypothesis class $\mathcal{F}$, the goal is to minimize the *task risk* over $f \in \mathcal{F}$

$$R(f) := \mathbb{E}_P \left[ \ell(\mathsf{d} \circ f(X), Y) \right]. \tag{1}$$

For example, in binary classification, $d = 1$, $\mathsf{d}(s) = \mathrm{sgn}(s)$, and $\ell(y, y') = 1\{yy' \leq 0\}$, yielding $R(f) = \mathbb{P}(Y f(X) \leq 0)$. The challenge of minimizing $R(f)$ is that the task loss $\ell$ can be nonsmooth, nonconvex, and—even more—uninformative: the loss landscape of the 0-1 loss is flat almost everywhere. This makes even practical (e.g., first-order) optimization impossible. We will consider a slightly more sophisticated version of the problem (1), where instead of the loss $\ell$ being defined only in terms of the instantaneous label $Y$, we will allow it to depend on $P(Y \in \cdot \mid X)$, so that we investigate

$$R(f) := \mathbb{E}_P \left[ \ell(\mathsf{d} \circ f(X), P(\cdot \mid X)) \right], \tag{2}$$

whose minimizers frequently coincide with the original problem (2), but which allows more sophistication. (For example, in multiclass classification, $Y \in \{1, \ldots, k\}$, and taking $\ell(\widehat{y}, P) = \sum_y P(Y = y) 1\{\widehat{y} \neq y\}$, the risk coincides with the standard 0-1 error rate.)

Instead of the task loss $\ell$, we thus consider an easier to optimize surrogate $\varphi : \mathbb{R}^d \times \mathcal{Y} \to \mathbb{R}$. Then rather than attacking the risk (2) directly, we minimize surrogate risk

$$R_\varphi(f) := \mathbb{E}_P \left[ \varphi(f(X), Y) \right].$$

For this to be sensible, we must exhibit some type of consistency with the task problem (2). In this paper, we particularly study in two scenarios, which we will make more formal:

(i) The "classical" case of Fisher consistency, where $\mathcal{F}$ contains all Borel functions;

(ii) Statistical scenarios in which the hypothesis class $\mathcal{F}$ is parametric but may be mis-specified.

Our main message is that label aggregation improves consistency in both scenarios, demonstrating the robustness of label cleaning.

### 2.1 Label aggregation

Instead of obtaining $(X_i, Y_i)$ pairs, consider the case that we replace the output $Y$ with a more abstract variable $Z \in \mathcal{Z}$. For example, in the motivating scenario in the introduction in which we collect multiple (say, $m$) noisy labels for each example $X$, we take $Z = (Y_1, \ldots, Y_m) \in \mathcal{Y}^m$. For an abstract "aggregation space" $\mathcal{A}$, let $A : \mathcal{Z} \to \mathcal{A}$ be an aggregating function (e.g., majority vote), and let $\varphi : \mathbb{R}^d \times \mathcal{A} \to \mathbb{R}_+$ be a surrogate loss defined on this aggregation space. We then define the aggregated surrogate risk

$$R_{\varphi, A}(f) := \mathbb{E} \left[ \varphi(f(X), A(Z)) \right], \tag{3}$$

asking when minimizing the surrogate problem (3) is sufficient to minimize the actual task risk (2). Two concrete examples may make this clearer.

**Example 1** (Majority vote): In the repeated sampling regime, data collection takes the form $Z = (Y_1, \cdots, Y_m)$, $Y_i \mid X = x \overset{\text{iid}}{\sim} P_{Y|X=x}$. Define $A_m(Z)$ to be the empirical minimizer

$$A(Z) = A(\{Y_1, \ldots, Y_m\}) = \operatorname*{argmin}_{y \in \mathcal{Y}} \sum_{l=1}^{m} \ell(y, Y_l).$$

When $\ell(\widehat{y}, y) = 1\{y \neq \widehat{y}\}$, this corresponds exactly to majority vote; the more general form allows more abstract procedures. ◊

We can also (roughly) capture $K$-nearest neighbor aggregation procedures:

**Example 2** ($K$-nearest neighbors): Consider an abstract repeated sampling scenario in which an example $X$ comes with a label $Y$ and an additional draw $(X_i, Y_i)_{i=1}^{m} \overset{\text{iid}}{\sim} P$, where $m$ is the number of additional examples, so $Z = (Y, (X_i, Y_i)_{i=1}^{m})$. Let $\operatorname{dist} : \mathcal{X} \times \mathcal{X} \to \mathbb{R}_+$ be a distance metric on $\mathcal{X}$. Let $\{X_{(1)}, \ldots, X_{(m)}\}$ order the input sample $\{X_i\}_{i=1}^{m}$ by distance, $\operatorname{dist}(X, X_{(1)}) \leq \ldots \leq \operatorname{dist}(X, X_{(m)})$ (and let $X_{(0)} = X$). For $K \geq 0$, we can aggregate the $K$-nearest neighbors of $X$, for example, by choosing

$$A_{m,K}(Z) \coloneqq \operatorname*{argmin}_{y \in \mathcal{Y}} \sum_{l=0}^{K} \ell(y, Y_{(l)}).$$

In Appendix E, we leverage the results in the coming sections to move beyond this population-level scenario to address aggregation from a single sample $(X_i, Y_i)_{i=1}^{n}$. ◊

## 3 SURROGATE CONSISTENCY

The standard framework for surrogate consistency Steinwart (2007) assumes that $\mathcal{F}$ consists of all Borel measurable functions $f : \mathcal{X} \to \mathbb{R}^d$. Working in the abstract setting in the preliminaries, define the conditional task risk $R(s \mid x)$ and the conditional surrogate risk $R_\varphi(s \mid x)$, $s \in \mathbb{R}^d$ by

$$R(s \mid x) \coloneqq \ell(\mathsf{d} \circ s, P(Y \in \cdot \mid X = x)) \text{ and } R_\varphi(s \mid x) \coloneqq \mathbb{E}\left[\varphi(s, Y) \mid X = x\right].$$

We then define the pointwise excess risks

$$\delta_\ell(s, x) \coloneqq R(s \mid x) - \inf_{s' \in \mathbb{R}^d} R(s' \mid x), \qquad \delta_\varphi(s, x) \coloneqq R_\varphi(s \mid x) - \inf_{s' \in \mathbb{R}^d} R_\varphi(s' \mid x),$$

as well as the minimal risks $R^\star \coloneqq \inf_{f \in \mathcal{F}} R(f)$ and $R_\varphi^\star \coloneqq \inf_{f \in \mathcal{F}} R_\varphi(f)$. We follow the standard Steinwart (2007); Bartlett et al. (2006); Zhang (2004c) that consistency requires at least (i) Fisher consistency and, if possible, a stronger and quantitative (ii) uniform comparison inequality: respectively, that for all data distributions $P$,

(i) For any sequence of functions $f_n \in \mathcal{F}$, $R_\varphi(f_n) \to R_\varphi^\star$ implies $R(f_n) \to R^\star$.

(ii) For a non-decreasing $\psi : \mathbb{R}_+ \to \mathbb{R}_+$, $\psi(R(f) - R^\star) \leq R_\varphi(f) - R_\varphi^\star$ for all $f \in \mathcal{F}$, where $\psi$ satisfies $\psi(\epsilon) > 0$ for all $\epsilon > 0$,

In the case of binary classification when $\varphi$ is margin-based and convex, the two consistency notions coincide (Bartlett et al., 2006). The stronger uniform guarantee (ii) need not always hold, the *calibration function* $\overline{\psi}$ provides a canonical construction through the excess risk:

$$\psi(\epsilon, x) \coloneqq \inf_{s \in \mathbb{R}^d} \{\delta_\varphi(s, x) \mid \delta_\ell(s, x) \geq \epsilon\} \text{ and } \overline{\psi}(\epsilon) \coloneqq \inf_{x \in \mathcal{X}} \psi(\epsilon, x).$$

Consistency and comparison inequalities follow from the calibration functions (see (Zhang, 2004c, Prop. 25) and (Steinwart, 2007, Thm. 2.8 and Lemma 2.9)):

**Corollary 3.1.** *The surrogate $\varphi$ is Fisher consistent* (i) *for $\ell$ if and only if $\psi(\epsilon, x) > 0$ for all $x \in \mathcal{X}$ and $\epsilon > 0$. Let $\psi$ be the Fenchel biconjugate of $\overline{\psi}$. Then $\overline{\psi}(\epsilon) > 0$ if and only if $\psi(\epsilon) > 0$, and for all measurable $f$,*

$$\psi(R(f) - R^\star) \leq R_\varphi(f) - R_\varphi^\star.$$

In the general risk minimization problem (2) we would like at least a Fisher-consistent (i) surrogate for $\ell$, so that minimizing $R_\varphi(f) = \mathbb{E}[\varphi(f(X), Y)]$ would imply minimizing $R(f)$. Given such a result, using only paired observations $(X, Y)$ rather than tuples $(X, Y_1, \ldots, Y_m)$, we could bring the entire theory of empirical processes and related statistical tools (van der Vaart and Wellner, 1996; Bartlett et al., 2005; Koltchinskii, 2006; Bartlett et al., 2006) to bear on the problem. Moreover, data collection procedures would be simpler, necessitating only single pairs $(X, Y)$ for consistent estimation. Unfortunately, such results are generally impossible, as we detail in the next extended example, necessitating the necessity of a theory of aggregation that we pursue in Sec. 3.2.

## 3.1 FISHER CONSISTENCY FAILURE WITHOUT LABEL AGGREGATION: RANKING

Consider the problem of ranking $k$ items using pairwise comparison data (Keener, 1993; Dwork et al., 2001; Duchi et al., 2012; Negahban et al., 2016), where the space $\mathcal{Y}$ consists of all pairwise comparisons of these items, $\mathcal{Y} = \{(i, j) : i \neq j, 1 \leq i, j \leq k\}$. The (population) rank aggregation problem is, for each $x$, to transform the probabilities $p_{ij} = P(Y = (i, j) \mid x)$ into a ranking of the $k$ items. While numerous possibilities exist for such aggregation, we consider a simple comparison-based aggregation scheme (cf. Keener, 1993); similar negative results to the one we show below hold for more sophisticated schemes. Define the normalized transition matrix $C_x \in \mathbb{R}_+^{k \times k}$ with entries $(C_x)_{ii} = 0$ and

$$(C_x)_{ij} = \frac{p_{ij}}{\sum_{l \neq j} p_{lj}} \ \text{ for } i \neq j,$$

where we let $0/0 = 1/(k-1)$ so that $C_x$ is stochastic, satisfying $C_x^T \mathbf{1} = \mathbf{1}$. We then rank the items by the vector $C_x \mathbf{1} \in \mathbb{R}_+^k$, which measures how often a given item is preferred to others. (One may also take higher powers $C_x^p \mathbf{1}$ or Perron vectors (Keener, 1993); similar results to ours below hold in such cases.) Tacitly incorporating the decoding d into the task loss $\ell$, we

$$\ell(s, C) := \max_{i < j} \mathbb{1}\left\{(s_i - s_j)(e_i - e_j)^T C \mathbf{1} \leq 0, \ (e_i - e_j)^T C \mathbf{1} \neq 0\right\},$$

which penalizes mis-ordered scores between $s$ and $C$. The population task risk (2) is thus

$$R(f) := \mathbb{P}\left(f(X) \text{ and } C_X \mathbf{1} \text{ order differently}\right) \tag{4}$$

Now consider a convex surrogate $\varphi : \mathbb{R}^k \times \mathcal{Y} \to \mathbb{R}$. We restrict to $s \in \mathbb{R}^k$ for which $s^T \mathbf{1} = 0$, a minor restriction familiar from multiclass classification problems (Zhang, 2004b; Tewari and Bartlett, 2007), which is natural as for decoding a ranking we require only the ordering of the $s_i$. Unfortunately, there is no convex Fisher-consistent surrogate for the problem (4) (see Appendix C.1).

**Proposition 1.** *Consider the ranking problem with task risk* (4) *over $k \geq 3$ outcomes. If $\varphi : \mathbb{R}^k \times \mathcal{Y} \to \mathbb{R}$ is convex in its first argument, it is not Fisher consistent.*

Nonetheless, a reasonably straightforward argument yields consistency when we allow aggregation methods as soon as $m$, the number of collected comparisons, satisfies $m \geq k$. The idea is simple: we regress predicted scores $f(x)$ on frequencies of label orderings. We assume multiple independent pairwise comparisons $Z = (Y_1, \ldots, Y_m)$ conditioned on $X$, and letting $m_{ij} = \sum_{y \in \mathcal{Y}^m} \mathbb{1}\{y = (i, j)\}$ and $m_j = \sum_{i=1}^k m_{ij}$, we define the aggregation

$$A(Z) = \begin{cases} \star, & \text{if } m_j = 0 \text{ for some } j \text{ in } [k], \\ \left(\frac{m_{i1}}{m_1} + \frac{m_{i2}}{m_2} + \cdots + \frac{m_{ik}}{m_k}\right)_{i \in [k]} & \text{otherwise, i.e. if } m_j > 0 \text{ for all } j \in [k]. \end{cases}$$

Regressing directly on $A(Z)$ when $A(Z) \neq \star$ yields consistency, as the next proposition demonstrates (see Appendix C.2 for a proof):

**Proposition 2.** *Define $\varphi(s, q) = \|s - q\|_2^2$ for $s, q \in \mathbb{R}^k$ and $\varphi(s, \star) = 0$. Then if $m \geq k$, $\varphi$ is Fisher consistent for the ranking risk* (4).

## 3.2 LABEL AGGREGATION OBTAINS STRONGER SURROGATE CONSISTENCY

The extended ranking example suggests potential benefits of aggregating labels, and it is natural to ask how aggregation interacts with surrogate consistency more generally. Thus, we present two results here: one that performs an essentially basic extension of standard surrogate-risk consistency, and the second that shows how aggregation-based methods can "upgrade" what might nominally be inconsistent losses into consistent losses, as Proposition 2 suggests may be possbile.

### 3.2.1 Basic Extensions of Surrogate Consistency

We begin by making the more or less obvious generalization of calibration functions for standard cases, extending the classical comparison inequalities in Corollary 3.1. For an arbitrary aggregation method $A : \mathcal{Z} \to \mathcal{A}$, define the conditional surrogate risk with data aggregation

$$R_{\varphi,A}(s \mid x) := \mathbb{E}\left[\varphi(s, A(Z)) \mid X = x\right].$$

As in the non-aggregated case, the pointwise excess risk

$$\delta_{\varphi,A}(s,x) := R_{\varphi,A}(s \mid x) - \inf_{s \in \mathbb{R}^d} R_{\varphi,A}(s \mid x)$$

then defines the pointwise and uniform calibration functions

$$\overline{\psi}_A(\epsilon, x) := \inf_{s \in \mathbb{R}^d} \{\delta_{\varphi,A}(s,x) \mid \delta_\ell(s,x) \geq \epsilon\} \ \text{ and } \ \overline{\psi}_A(\epsilon) := \inf_{s \in \mathbb{R}^d} \overline{\psi}_A(\epsilon, x). \tag{5}$$

A consistency result then follows, similar to Corollary 3.1, under appropriate measurability conditions (we will leave these tacit as they are not central to our results). Then more or less as a corollary of Steinwart (2007, Thm. 2.8), we have the following consistency result. (We include a proof for completeness in Appendix D.1.)

**Proposition 3.** *Assume there exists $b : \mathcal{X} \to \mathbb{R}_+$ with $\int b(x)dP(x) < \infty$ such that $\delta_\ell(f(x),x) \leq b(x)$. The surrogate $\varphi$ is Fisher consistent* (i) *for the task risk* (2) *if and only if $\overline{\psi}_A(\epsilon, x) > 0$ for all $x \in \mathcal{X}$ and $\epsilon > 0$. Additionally, if $\psi_A = (\overline{\psi}_A)^{**}$ is the Fenchel biconjugate of $\overline{\psi}_A$, then*

$$\psi_A(R(f) - R^\star) \leq R_{\varphi,A}(f) - R^\star_{\varphi,A}.$$

The result captures the classical consistency guarantees—nothing particularly falls apart because of aggregation—but it provides no specific guarantees of improved consistency. We turn to this now.

### 3.2.2 Identifying Surrogates and Consistency

We now turn under essentially minimal conditions on the surrogate, there is a generic aggregating strategy that (asymptotically in the number of observations $y$) guarantees consistency for any task loss that seeks to minimize $\ell(f(x), y)$, i.e., $R(f) = \mathbb{E}[\ell(f(X), Y)]$. We assume that $\text{card}(\mathcal{Y}) = k < \infty$, and we impose a minimal identifiability assumption on the surrogate loss.

**Definition 3.1** (Identifying surrogate). *A surrogate $\varphi : \mathbb{R}^d \times \mathcal{A} \to \mathbb{R}$ is $(\mathsf{C}_{\varphi,1}, \mathsf{C}_{\varphi,2})$-identifying for $\mathcal{Y}$, $0 < \mathsf{C}_{\varphi,1} \leq \mathsf{C}_{\varphi,2} < \infty$ if there exist $\{a_y\}_{y \in \mathcal{Y}} \subset \mathcal{A}$ and vectors $\{s_y\}_{y \in \mathcal{Y}}$ such that $\mathsf{d}(s_y) = y$ and for which for all $y \neq y'$,*

$$\varphi(s_y, a_y) + \mathsf{C}_{\varphi,1} \leq \inf_{\mathsf{d} \circ s \neq y} \varphi(s, a_y), \tag{6a}$$

$$\varphi(s_y, a_{y'}) - \mathsf{C}_{\varphi,2} \leq \inf_{s \in \mathbb{R}^d} \varphi(s, a_{y'}). \tag{6b}$$

Inequality (6a) captures that for each class $y \in \mathcal{Y}$, there exists a parameter $a_y \in \mathcal{A}$ such that the minimizer of $\varphi(\cdot, a_y)$ identifies $y$. A finite $\mathsf{C}_{\varphi,2}$ exists for (6b) if $\varphi(\cdot, a)$ has a finite lower bound. Notably, Definition 3.1 does not require that $\varphi(\cdot, a)$ is convex or that it is consistent when $\mathcal{A} = \mathcal{Y}$ and $\mathcal{Z} = Y$, i.e., without label aggregation.

**Example 3:** Consider the binary hinge loss $\varphi(s, a) = \max\{1 - sa, 0\}$ for $\mathcal{A} = \mathcal{Y} = \{\pm 1\}$. For $y \in \{-1, 1\}$, take $a_y = s_y = y$, so that $\varphi(s_1, a_1) = \varphi(s_{-1}, a_{-1}) = 0$, while $\inf_{sa \leq 0} \varphi(s, a) = 1$. Similarly, $\varphi(s_1, a_{-1}) = \varphi(s_{-1}, a_1) = 2$, so the hinge loss is $(1, 2)$-identifying. $\Diamond$

Given an identifying surrogate with parameters $\{a_y\}_{y \in \mathcal{Y}}$, we consider a naive aggregation strategy: the generalized majority vote

$$A_m(y_1, \ldots, y_m) := a_{\hat{y}} \ \text{ for } \ \hat{y} = \operatorname*{argmin}_{y \in \mathcal{Y}} \sum_{i=1}^{m} \ell(y, y_i) \tag{7}$$

(breaking ties arbitrarily). As $m \to \infty$, because $\mathcal{Y}$ is finite, whenever $Y_i$ are i.i.d. there necessarily exists a (random) $M < \infty$ such that $m \geq M$ implies

$$\operatorname*{argmin}_{y \in \mathcal{Y}} \left\{ \sum_{i=1}^{m} \ell(y, Y_i) \right\} \subset y^\star(x) := \operatorname*{argmin}_{y \in \mathcal{Y}} \mathbb{E}\left[\ell(y, Y) \mid X = x\right].$$

From this, we expect that as $m \to \infty$, the surrogate $\varphi(\cdot, A_m)$ ought to be consistent. In fact, we have the following corollary of our coming results, guaranteeing (asymptotic) consistency:

**Corollary 3.2.** *Let $m = m(n) \to \infty$ and $\varphi$ be identifying (Def. 3.1). Then*

$$R_{\varphi, A_m}(f_n) - R^\star_{\varphi, A_m} \to 0 \;\; implies \;\; R(f_n) - R^\star \to 0.$$

### 3.2.3 IDENTIFYING SURROGATES AND CONSISTENCY AMPLIFICATION

In cases with low noise in the labels, the aggregation strategy (7) allows an explicitly improved comparison inequality $\psi(R(f) - R^\star) \le R_\varphi(f) - R^\star$, in that $\psi$ is linear over some range of $\epsilon > 0$—and linear growth is the strongest comparison inequality possible Osokin et al. (2017); Nowak-Vila et al. (2020). More generally, strict comparison inequalities, such as those present in Proposition 3, can be too narrow, as it can still be practically convenient to adopt inconsistent surrogates (Liu, 2007; Osokin et al., 2017; Nowak-Vila et al., 2020). Thus, we follow Osokin et al. (2017) to introduce $(\xi, \zeta)$ *consistency*, which requires a comparison function $\psi$ to grow linearly only for $\epsilon \ge \xi$, so that the surrogate captures a sort of "good enough" risk.

**Definition 3.2.** *The surrogate loss $\varphi$ and aggregator $A$ yield* level-$(\xi, \zeta)$ consistency *if there exists $\psi$ satisfying $\psi(\epsilon) \ge \zeta\epsilon$ for $\epsilon \ge \xi$, and $\psi(R(f) - R^\star) \le R_{\varphi,A}(f) - R^\star_{\varphi,A}$.*

In the following discussion, we show under minimal assumptions, label aggregation (7) can achieve level-$(o_m(1), \zeta)$ consistency even if the surrogate is Fisher inconsistent.

We introduce a quantifiable noise condition, adapting the now classical Mammen-Tsybakov noise conditions Mammen and Tsybakov (1999) (see also Bartlett et al. (2006)). Define

$$\Delta(x) \coloneqq \min_{\mathsf{d}(s) \notin y^\star(x)} \delta_\ell(s, x), \tag{8}$$

the minimal excess conditional risk when making an incorrect prediction. In binary classification problems with $\mathcal{Y} = \{\pm 1\}$, one obtains $\Delta(x) = |2P(Y = 1 \mid X = x) - 1|$, and more generally, we expect that consistent estimation should be harder when $\Delta(x)$ is closer to 0. We can define the Mammen-Tsybakov conditions (where the constant $\mathsf{C}_{\mathsf{MT}} > 0$ may change) as

$$\mathbb{P}\left(\mathsf{d} \circ f \neq \mathsf{d} \circ f^\star\right) \le \mathsf{C}_{\mathsf{MT}} \left(R(f) - R^\star\right)^\alpha \;\; \text{for all measurable } f, \tag{$N_\alpha$}$$

where we refer to condition ($N_\alpha$) as having *noise exponent* $\alpha$, and

$$\mathbb{P}(\Delta(X) \le \epsilon) \le (\mathsf{C}_{\mathsf{MT}}\epsilon)^\beta \;\; \text{for } \epsilon > 0. \tag{$M_\beta$}$$

Here, $\alpha \in [0, 1]$ and $\beta \in [0, \infty]$, so that conditions ($N_\alpha$) and ($M_\beta$) always trivially hold with $\alpha = \beta = 0$, moreover, as in the binary case (Bartlett et al., 2006, Thm. 3), they are equivalent via the transformation $\beta = \frac{\alpha}{1-\alpha}$. (See Appendix D.2.) We shall also use a *noise condition number*

$$\kappa(x) \coloneqq \frac{\max_{\mathsf{d}(s) \neq y^\star(x)} \delta_\ell(s, x)}{\min_{\mathsf{d}(s) \neq y^\star(x)} \delta_\ell(s, x)}, \tag{9}$$

which connects the noise statistic $\Delta(x)$ and the pointwise excess risk via $\Delta(x) \ge \delta_\ell(s, x)/\kappa(x)$ for all $s$ such that $\mathsf{d}(s) \neq y^\star(x)$, allowing more fine-grained analysis. In binary classification, we have $\kappa(x) = 1$ so long as $\mathbb{P}(\Delta(X) > 0) = 1$.

The noise statistic $\Delta(x)$ and condition number $\kappa(x)$ will allow us to show how (generalized) majority vote (7), when applied in the context of any identifiable surrogate (Definition 3.1), achieves level-$(\xi, \zeta)$ consistency. Define the error function

$$e_m(t) \coloneqq t\sqrt{\frac{2}{m} \log\left(\frac{4k(\mathsf{C}_{\varphi,1} + \mathsf{C}_{\varphi,2})}{\mathsf{C}_{\varphi,1}}\right)}, \tag{10}$$

which roughly captures that if $\kappa(x) = t$, then majority vote $A_m$ is likely correct if $m$ is large enough that $e_m(t) \ll 1$. We then have the following theorem, which provides a (near) linear calibration function; we prove it in Appendix D.3.

**Theorem 1.** *Let the surrogate loss $\varphi$ be $(\mathsf{C}_{\varphi,1}, \mathsf{C}_{\varphi,2})$-identifying with parameters $\{a_y\}_{y \in \mathcal{Y}}$, and $A_m$ be the majority vote aggregator (7). Assume the task loss satisfies $0 \le \ell \le 1$ and $P$ satisfies condition ($N_\alpha$) with noise exponent $\alpha \in [0, 1]$. Then for any $M > 0$ and $f \in \mathcal{F}$ such that $R(f) - R^\star \ge 2\mathbb{P}(\kappa(X) > M) + (4\mathsf{C}_{\mathsf{MT}}e_m(M))^{\frac{1}{1-\alpha}}$,*

$$R(f) - R^\star \le \frac{16}{\mathsf{C}_{\varphi,1}} \cdot \left(R_{\varphi, A_m}(f) - R^\star_{\varphi, A_m}\right).$$

Said differently, under the conditions of the theorem, $\varphi$ with aggregation provides level $(\xi, \zeta)$ consistency (Def. 3.2) with $\xi = 2\mathbb{P}(\kappa(X) > M) + (4\mathsf{C}_{\mathsf{MT}}e_m(M))^{\frac{1}{1-\alpha}}$ and $\zeta = \frac{\mathsf{C}_{\varphi,1}}{16}$. Theorem 1 also provides an immediate proof of Corollary 3.2, that is, an asymptotic guarantee of consistency. Indeed, define

$$\xi_m := \inf_M \left\{ 2\mathbb{P}(\kappa(X) > M) + (4\mathsf{C}_{\mathsf{MT}}e_m(M))^{\frac{1}{1-\alpha}} \right\},$$

which satisfies $\xi_m \to 0$ as $m \to \infty$, because $\mathbb{P}(\kappa(X) > M) \to 0$ as $M \uparrow \infty$ and for any fixed $M$, $e_m(M) \to 0$ as $m$ grows. Corollary 3.2 then follows trivially by taking $\alpha = 0$.

Theorem 1 is a somewhat gross result, as the identifiability conditions in Def. 3.1 are so weak. With a tighter connection between task loss $\ell$ and surrogate $\varphi$, for example, making the naive majority vote (7) more likely to be correct (or at least correct enough for $\varphi$), we would expect stronger bounds. We do not pursue this here.

To provide a somewhat more concrete bound, we optimize over $M$ in Theorem 1, using the crued bound $\kappa(x) \le 1/\Delta(x)$ on the condition number. By taking $M = 1$ for $\mathrm{card}(\mathcal{Y}) = k = 2$ and optimizing $M$ for $k \ge 3$, we may lower bound $\xi_{m,k}$ in the level $(\xi_{m,k}, \zeta)$-consistency (Def. 3.2) that in Theorem 1 promises, setting

$$\xi_{m,k} := \begin{cases} \left( \frac{32\mathsf{C}_{\mathsf{MT}}^2}{m} \log\left( \frac{8(\mathsf{C}_{\varphi,1}+\mathsf{C}_{\varphi,2})}{\mathsf{C}_{\varphi,1}} \right) \right)^{\frac{1}{2(1-\alpha)}}, & \text{if } k = 2 \\ 4 \cdot \left( \frac{32\mathsf{C}_{\mathsf{MT}}^4}{m} \log\left( \frac{4k(\mathsf{C}_{\varphi,1}+\mathsf{C}_{\varphi,2})}{\mathsf{C}_{\varphi,1}} \right) \right)^{\frac{\alpha}{2(1-\alpha^2)}}, & \text{otherwise.} \end{cases}$$

Making appropriate algebraic substitutions and manipulations (see Appendix D.4), we have the following corollary.

**Corollary 3.3.** *Under the conditions of Theorem 1, for any $f$ such that $R(f) - R^\star \ge \xi_{m,k}$,*

$$R(f) - R^\star \le \frac{16}{\mathsf{C}_{\varphi,1}} \cdot \left( R_{\varphi,A_m}(f) - R_{\varphi,A_m}^\star \right).$$

The above corollary and Corollary 3.2 provide evidence for the robustness of label cleaning: with minimal assumptions on the surrogate, data aggregation can still yield consistency. As the noise exponent $\alpha$ approaches 1 in Corollary 3.3, the sample size $m$ required for the comparison inequality to hold for a fixed score function $f$ shrinks. Notably, if $\alpha = 1$, whenever

$$m \ge 32 \max\{\mathsf{C}_{\mathsf{MT}}^2, \mathsf{C}_{\mathsf{MT}}^4\} \cdot \log\left( \frac{4k(\mathsf{C}_{\varphi,1} + \mathsf{C}_{\varphi,2})}{\mathsf{C}_{\varphi,1}} \right) = O(\log k),$$

we have $\xi_{m,k} = 0$, yielding the uniform comparison inequality (ii) with linear comparison. The noise level of the learning problem itself affects the aggregation level needed for consistency—an "easier" problem requires less aggregation to achieve stronger consistency.

We collect several examples in Appendix A, of varying levels of concreteness, that allow us to instatiate Theorem 1 and Corollary 3.3. Those examples are (i) binary classification with a nonsmooth surrogate, (ii) bipartite matching and (iii) structured prediction.

## 4 ROBUSTNESS AND CONSISTENCY FOR MODELS

The previous section builds off of the now classical theory of surrogate risk consistency, which assumes $\mathcal{F}$ to be the class of all measurable functions. The results there show that aggregation can allow us to "upgrade" consistency so that even if a surrogate $\varphi$ is inconsistent for paired (non-aggregated) data $(X, Y)$, we can achieve level-$(\xi, \zeta)$ consistency (Def. 3.2) with sufficient aggregation. Here, we take a different view of the problem of consistency, considering the consequences of optimizing over a restricted (often parametric) hypothesis class $\mathcal{F}$. Of course, in a well-specified model, obtaining consistency with such a restricted hypothesis class is no issue, but it is unrealistic to assume such a brittle condition. This gives rise to the long-standing challenge of quantifying surrogate consistency when the hypothesis class contains only a subset of the measurable functions (Duchi et al., 2016; Nguyen et al., 2009). We tackle some of the issues around this, showing that aggregating labels allows consistent estimates in scenarios where consistency might otherwise fail. We identify one such failure mode for binary classification in restricted class and show how aggregation can generically achieve consistency. We postpone details to Proposition 5 and Theorem 3 in Appendix B.

## 4.1 ON FINITE-DIMENSIONAL MULTICLASS CLASSIFICATION

The final technical content of this paper considers a multiclass scenario in which $\mathcal{X} = \mathbb{R}^d$, $\mathcal{Y} = [k]$, and we use linear predictors, but where the predictive model may be mis-specified. We will show that even when the problem is optimally predictable and the linear model is well-specified on all except an $\epsilon$-fraction of data, surrogate risks based only on $(X, Y)$ pairs are inconsistent; majority vote-based methods, however, will recover the optimal linear predictor.

To set the stage, let $\Theta = [\theta_1 \cdots \theta_{k-1}] \in \mathbb{R}^{d \times k-1}$, and let the labels follow a categorical distribution $Y \mid X = x \sim \mathsf{Cat}(p_\Theta(x))$, where $p_\Theta(x) \in \mathbb{R}_+^k$ satisfies $\mathbf{1}^\top p_\Theta(x) = 1$ and

$$p_\Theta(x) = \begin{bmatrix} \sigma(\langle \theta_1, x \rangle, \cdots, \langle \theta_{k-1}, x \rangle) \\ 1 - \mathbf{1}^\top \sigma(\langle \theta_1, x \rangle, \cdots, \langle \theta_{k-1}, x \rangle) \end{bmatrix}$$

for a link $\sigma : \mathbb{R}^{k-1} \to \mathbb{R}_+^{k-1}$, $\mathbf{1}^\top \sigma \le 1$. We assume $\sigma$ satisfies the consistency condition that for $t \in \mathbb{R}^{k-1}$, setting $t_k = 0$ and $\sigma_k(t) = 1 - \mathbf{1}^\top \sigma(t)$,

$$t_y = \max_{1 \le i \le k} t_i \text{ if and only if } \sigma_y(t) = \max_{1 \le i \le k} \sigma_i(t). \tag{11}$$

One standard example is multiclass logistic regression, where $\sigma_i^{\mathrm{lr}}(t) = \frac{e^{t_i}}{1 + e^{t_1} + \cdots + e^{t_{k-1}}}$. Let the multilabeled dataset be $\{(X_i; (Y_{1i}, \cdots, Y_{im}))\}_{i=1}^n$ with repeated sampling $Y_{ij} \mid X_i \overset{\text{iid}}{\sim} \mathsf{Cat}(p_{\Theta^\star}(X_i))$, and $Y_i^+ = A_m(Y_{i1}, \cdots, Y_{im})$ be the majority vote with ties broken arbitrarily. Then $Y_i^+ \mid X_i \sim \mathsf{Cat}(p_{\Theta^\star, m}(X_i))$, where if $\rho_m(t)$ denotes the distribution of majority vote on $m$ items with initial probabilities $\sigma(t) \in \mathbb{R}_+^{k-1}$, then $p_{\Theta, m}(x) = \begin{bmatrix} \rho_m(\langle \theta_1, x \rangle, \cdots, \langle \theta_{k-1}, x \rangle) \\ 1 - \mathbf{1}^\top \rho_m(\langle \theta_1, x \rangle, \cdots, \langle \theta_{k-1}, x \rangle) \end{bmatrix}$.

It is evident that $\rho_m$ satisfies link consistency (11). Consider fitting a logistic regression with loss $\varphi(\Theta^\top x, y) = -\langle \theta_y, x \rangle + \log \left( 1 + \sum_{i=1}^{k-1} \exp(\langle \theta_i, x \rangle) \right)$, with the convention that $\theta_k = \mathbf{0}$, and let $L_m(\Theta) = \mathbb{E}[\varphi(\Theta^\top X, Y_m^+)]$ be the logistic loss with $m$-majority vote. Then

$$\nabla_{\theta_i} \varphi(\Theta^\top x, y) = -x(1\{y = i\} - \sigma_i^{\mathrm{lr}}(\Theta^\top x)),$$

so that the $\Theta_m$ minimizing $L_m$ satisfies

$$\nabla_\Theta L_m(\Theta_m) = \mathbb{E} \left[ X \left( \sigma^{\mathrm{lr}}(\Theta_m^\top X) - \rho_m(\Theta^{\star \top} X) \right)^\top \right] = 0.$$

Standard results in statistics guarantees both consistency and efficiency when the model is well-specified without aggregation, and when $m \ge 1$ and $k = 2$, Cheng et al. (2022, Prop. 3) show there exists $t_m > 0$ such that $\Theta_m = t_m \Theta^\star$ if $X \sim \mathsf{N}(0, I_d)$ even with a mis-specified link. This implies that in binary classification, even if the link function is incorrect, we can still achieve consistent classification regardless of the aggregation level $m$, as the direction $\Theta^\star / \|\Theta^\star\|_2$ determines consistency. However, as soon as $k \ge 3$ and the true link is slightly mis-specified, risk consistency fails. Fixing a set $\mathcal{T}_\epsilon \subset \mathbb{R}^{k-1}$ with Lebesgue measure $\epsilon$, consider

$$\sigma^\epsilon(t) = \sigma^{\mathrm{lr}}(t) 1\{t \notin \mathcal{T}_\epsilon\} + \frac{1}{k} \mathbf{1} \cdot 1\{t \in \mathcal{T}_\epsilon\},$$

which defines a distribution on $Y \in \{1, \ldots, k\}$ conditional on $t \in \mathbb{R}^{k-1}$ that samples $Y \sim \sigma^{\mathrm{lr}}(t)$ if $t \in \mathcal{T}_\epsilon$ and uniformly otherwise. Clearly $\sigma^\epsilon$ satisfies the link consistency condition (11) and is optimally predictable (13). For $\epsilon > 0$, define

$$L_{m,\epsilon}(\Theta) := \mathbb{E}[\mathbb{E}_{\sigma^\epsilon}[\varphi(\Theta^\top X, Y_m^+) \mid X]]$$

to be the (population) logistic loss, based on $m$-majority vote, when $Y \mid X = x \sim \sigma^\epsilon(\Theta^{\star \top} x)$. Let $\Theta_m(\epsilon) = \operatorname{argmin}_\Theta L_{m,\epsilon}(\Theta)$. Evidently, $\Theta_1(0) = \Theta^\star$; nonetheless, the next result shows that for arbitrarily small $\epsilon > 0$, consistency fails without aggregation. See its proof in Appendix G.1.

**Proposition 4.** *Let $k \ge 3$, $\Sigma = I$. Assume that for $Z \sim \mathsf{N}(0, I_{k-1})$, the linear mapping $M \mapsto DM := \mathbb{E}[Z(\nabla \sigma^{\mathrm{lr}}(\Theta^{\star \top} Z) M Z)^\top]$ is invertible. Then there exists $\epsilon_0 > 0$ such that for any $\epsilon \in (0, \epsilon_0)$, there is a set $\mathcal{T}_\epsilon$ with Lebesgue measure at most $\epsilon$ and for which*

$$\Theta_1(\epsilon) / \|\Theta_1(\epsilon)\| \ne \Theta^\star / \|\Theta^\star\|.$$

Majority vote, however, can address this inconsistency as $m \to \infty$ without any assumptions on the true link $\sigma$ except that it satisfies the consistency condition (11). Indeed, letting $L_{m,\sigma}(\Theta) = \mathbb{E}[\mathbb{E}_\sigma[\varphi(\Theta^\top X, Y_m^+) \mid X]]$ and $\Theta_m = \arg\min L_{m,\sigma}(\Theta)$, we have the following

**Theorem 2.** *Let $\Theta^\star$ have decomposition $\Theta^\star = U^\star T^\star$, where $U^\star \in \mathbb{R}^{d\times(k-1)}$ is orthogonal and $T^\star \in \mathbb{R}^{(k-1)\times(k-1)}$ is nonsingular. Then there exists $T_m \in \mathbb{R}^{(k-1)\times(k-1)}$ such that $\Theta_m = U^\star T_m$ for every $m$, and as $m \to \infty$,*

$$\|T_m\| \to \infty \ \ and \ \ T_m/\|T_m\| \to T^\star/\|T^\star\|.$$

See Appendix G.2 for a proof.

Theorem 2 shows more evidence for the robustness properties of label aggregation, providing asymptotic consistency even in mis-specified models so long as there is *some* link function describing the relationship between $X$ and $Y$. The robustness is striking when $k \geq 3$: as Proposition 4 highlights, methods without label aggregation are generally inconsistent.

## 5 DISCUSSION AND FUTURE WORK

The question of whether and how to clean data has animated much of the research discussion around dataset collection. Cheng, Asi, and Duchi (2022) provide a discussion of these issues, highlighting that there appears to be a phenomenon that using non-aggregated data—all available labels—leads to better statistical efficiency when models have the power to fully represent all uncertainty, but otherwise, data cleaning appears to be more robust. In a similar vein, Dorner and Hardt (2024) argue that, in a validation setting of comparing binary classifiers, it is better to use more noisy labels rather than cleaned variants. This paper contributes to this dialogue by providing evidence for both fundamental limits to using un-cleaned, un-aggregated label information in supervised learning while highlighting robustness improvements that come from label cleaning. Still, many questions remain.

**Finite $m$ results and fundamental limits.** Many of the consistency results we present repose on taking a limit as $m$, the number of labels aggregated, tends to infinity. While at some level, the purpose of this paper is to highlight ways in which label aggregation can improve robustness, it is perhaps unsatisfying to rely on this asymptotic setting. In the context of ranking (Sec. 3.1), we can provide explicit consistency guarantees at a finite $m$, but developing this further provides one of the most natural and (we believe) important avenues for future work. Providing a surrogate consistency theory that depends both on the loss pairs $(\ell, \varphi)$ and the available label count $m$ would be interesting; for example, in the context of ranking in Sec. 3.1, if we wish to look at second or third-order comparisons of items (e.g., powers $C_x^p$, as Keener (1993) suggests), do we require increasing label counts $m$? Precisely delineating those problems that require label cleaning and aggregation from those that do not represents a central challenge here.

**Fundamental limits of the noise condition number.** Our work relies on the noise condition number (9), $\kappa(X)$, to characterize comparison inequalities for multiclassification problems, hinting at the difficulty beyond binary classification, where trivially $\kappa(X) = 1$. The condition number can still be large even when the Mammen-Tsybakov noise level ($N_\alpha$) is low—i.e. $\alpha \approx 1$—in cases beyond binary classification. This is a consequence of the minimal assumptions on the surrogate in our setting, and it would be beneficial to identify connections between the loss $\ell$ and surrogate $\varphi$ that more closely capture problem difficulty. A more precise delineation of fundamental limits by constructing explicit failure modes will also yield more insights into fitting predictive models.

**Behavior in mis-specified models.** Our results on mis-specified models, especially those in Section 4.1, require optimal predictability (13), that is, that a Bayes-optimal classifier lie in $\mathcal{F}$. While classical surrogate consistency results provably fail even in this case—and methods based on aggregated labels can evidently succeed—moving beyond such restricted scenarios seems a fruitful and interesting direction. Nguyen et al. (2009), followed by Duchi et al. (2016), identify one direction here, showing that in binary and multiclass classification (respectively), jointly inferring a predictor $f$ and a data representation for $x$ requires that surrogates $\varphi$ take a particular form depending on the task loss $\ell$. These still repose on infinitely powerful decision rules $f$, however, so we need new approaches.

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

## A    SURROGATE CONSISTENCY EXAMPLES WITH MAJORITY VOTE

Throughout, we shall assume that $P$ has a noise exponent $\alpha \in [0, 1]$, though this is no loss of generality, as Condition ($N_\alpha$) always holds with $\alpha = 0$. We defer proofs for each result in this section to Appendix D.5.

### A.1    BINARY CLASSIFICATION WITH A NONSMOOTH SURROGATE

Consider the binary classification problem with a margin-based surrogate $\varphi(f(x), y) = \phi(yf(x))$, where $\phi$ is convex; Bartlett et al. (2006) show that $\varphi$ is consistent if and only if $\phi'(0) < 0$. Here, we show a (somewhat trivial) example for the robustness data aggregation offers by demonstrating that even if $\phi$ is inconsistent without aggregation, it can become so with it. Note, of course, that one would never *use* such a surrogate, so one ought to think of this as a thought experiment. Assume that the subgradient set $\partial\phi(0) \subset (-\infty, 0)$ and $\phi$ is convex with $\lim_{t \to \infty} \phi(t) = 0$.

**Lemma A.1.** *For any $\delta > 0$, $\varphi$ is $(\mathsf{C}_{\varphi,1}, \mathsf{C}_{\varphi,2})$-feasible with*

$$\mathsf{C}_{\varphi,1} = \phi(0) - \phi(\delta) > 0 \ \text{ and } \ \mathsf{C}_{\varphi,2} = \phi(-\delta).$$

Corollary 3.3 thus applies with $k = 2$, so if $f : \mathcal{X} \to \mathbb{R}$ satisfies

$$R(f) - R^\star \geq \left( \frac{32\mathsf{C}_{\mathsf{MT}}^2}{m} \log \left( \frac{8(\phi(-\delta) + \phi(0) - \phi(\delta))}{\phi(0) - \phi(\delta)} \right) \right)^{\frac{1}{2(1-\alpha)}},$$

then

$$R(f) - R^\star \leq \frac{16}{\phi(0) - \phi(\delta)} (R_{\varphi, A_m}(f) - R^\star_{\varphi, A_m}).$$

### A.2    BIPARTITE MATCHING

In general structured prediction problems (Nowozin and Lampert, 2011), an embedding map $v : \mathcal{Y} \to \mathbb{R}^d$ encodes structural information about elements $y \in \mathcal{Y}$, where $\mathcal{Y}$ is some "structured" space, which is typically large. Using decoder $\mathsf{d}(s) = \operatorname{argmax}_{y \in \mathcal{Y}} \langle s, v(y) \rangle$, for a loss $\ell : \mathcal{Y} \times \mathcal{Y} \to \mathbb{R}_+$ with $\ell(y, y) = 0$, the maximum-margin surrogate (generalized hinge loss) (Taskar et al., 2003; Tsochantaridis et al., 2004; Joachims, 2006) takes the form

$$\varphi(s, y) = \max_{\hat{y} \in \mathcal{Y}} \left( \ell(\hat{y}, y) + \langle v(\hat{y}) - v(y), s \rangle \right). \tag{12}$$

Notably, the loss (12) is typically inconsistent, except in certain low noise cases (Osokin et al., 2017; Nowak-Vila et al., 2020).

Before discussing structured prediction broadly, we consider bipartite matching. A bipartite matching consists of a graph $G = (V, E)$ where the vertices $V = V_1 \cup V_2$ partition into left and right sets $V_1 = \{1, \ldots, N\}$ and $V_2 = \{N + 1, \ldots, 2N\}$, while the $N$ edges $E$ each connect exactly one (unique) node in $V_1$ and $V_2$. Letting $\mathcal{Y}$ be the collection of all bipartite matching between $V_1$ and $V_2$, we evidently have $k = \operatorname{card}(\mathcal{Y}) = N!$. For any graph $G$, the embedding map

$$v(G) := (1\{(u, v) \in E\})_{u \in V_1, v \in V_2} \in \mathbb{R}^{N^2}$$

indexes edges, yielding $d = N^2$. The task loss counts the number of mistaken edges,

$$\ell(y_1, y_2) := \frac{1}{2N} \|v(y_1) - v(y_2)\|_1 = \frac{1}{2N} \|v(y_1) - v(y_2)\|_2^2.$$

In this case, the max-margin (structured hinge loss) surrogate (12) is identifying:

**Lemma A.2.** *For the bipartite matching problem on $2N$ vertices, the structured hinge loss* (12) *surrogate $\varphi$ is $(\mathsf{C}_{\varphi,1}, \mathsf{C}_{\varphi,2})$-identifying (Def. 3.1) with*

$$\mathsf{C}_{\varphi,1} = \frac{1}{N} \ \text{ and } \ \mathsf{C}_{\varphi,2} = 2.$$

The important consequence of Lemma A.2 is that even when $k = \text{card}(\mathcal{Y}) = N!$, aggregation-based methods can yield consistency (via the structured hinge loss) once $m$, the number of aggregated labels, exceeds $O(N \log N)$. As one specialization, substituting these constants into Corollary 3.3 for $k \geq 3$, for all measurable $f : \mathcal{X} \to \mathbb{R}^d$ such that

$$R(f) - R^\star \geq 4 \cdot \left( \frac{32 \mathsf{C}_{\mathsf{MT}}^4}{m} \log\left(4k(2N+1)\right) \right)^{\frac{\alpha}{2(1-\alpha^2)}},$$

one has

$$R(f) - R^\star \leq 16N(R_{\varphi, A_n}(f) - R^\star_{\varphi, A_n}).$$

### A.3 STRUCTURED PREDICTION

We return to the more general structured prediction setting, as at the beginning of the preceding subsection. Suppose the decoder d can pick any class $y \in \mathcal{Y}$, in that for each $y \in \mathcal{Y}$, the collection

$$\mathcal{S}(y) := \{s : \langle v(y), s \rangle > \langle v(\hat{y}), s \rangle, \text{for } \hat{y} \neq y \}$$

of selecting $s$ is non-empty. For each $y \in \mathcal{Y}$, define the identifiability gap

$$\tau(y) := \inf_{s \in \mathcal{S}(y)} \max_{y_+, y_- \neq y} \frac{\ell(y_+, y)}{\langle v(y) - v(y_+), s \rangle} \cdot \frac{\langle v(y) - v(y_-), s \rangle}{\ell(y_-, y)}.$$

We have the following identifiability guarantee.

**Lemma A.3.** *For the structured prediction problem, the max-margin* (12) *surrogate* $\varphi$ *is* $(\mathsf{C}_{\varphi,1}, \mathsf{C}_{\varphi,2})$-*identifiable with*

$$\mathsf{C}_{\varphi,1} = \min_{\hat{y} \neq y} \ell(\hat{y}, y), \qquad \mathsf{C}_{\varphi,2} = \max_{y \in \mathcal{Y}} \tau(y) + 1.$$

*In particular, if* $v(y) \in \{0, 1\}^d$ *and* $\ell(\hat{y}, y) = \frac{1}{2d} \|v(\hat{y}) - v(y)\|_1$, $\tau(y) = 1$ *for all* $y$ *and* $\mathsf{C}_{\varphi,2} = 2$.

Completing the example, as in the binary matching case, we see that nontrivial consistency guarantees hold once $m \geq \log \text{card}(\mathcal{Y})$. As $0 \leq \ell(\cdot, \cdot) \leq 1$, Corollary 3.3 applies, which yields for all measurable $f : \mathcal{X} \to \mathbb{R}^d$ that

$$R(f) - R^\star \geq \left( \frac{32 \mathsf{C}_{\mathsf{MT}}^4}{m} \log\left( 4|\mathcal{Y}| \left( 1 + \frac{\max_{\hat{y} \in \mathcal{Y}} \tau(y) + 1}{\min_{\hat{y} \neq y} \ell(\hat{y}, y)} \right) \right) \right)^{\frac{\alpha}{2(1-\alpha^2)}},$$

implies

$$R(f) - R^\star \leq \frac{8}{\min_{\hat{y} \neq y} \ell(\hat{y}, y)} (R_{\varphi, A_m}(f) - R^\star_{\varphi, A_m}).$$

## B CONSISTENCY AND AGGREGATION IN RESTRICTED HYPOTHESIS CLASS

### B.1 CONSISTENCY FAILURE FOR BINARY CLASSIFICATION IN FINITE DIMENSIONS

To see how restricting the hypothesis class can change the problem substantially even in well-understood cases, we consider binary classification. In this case, $\mathcal{Y} = \{\pm 1\}$, and we take the zero-one error $\ell(\mathsf{d}(s), y) = 1\{ys \leq 0\}$. We consider a margin-based surrogate $\varphi(s, y) = \phi(sy)$, where $\phi : \mathbb{R} \to \mathbb{R}_+$ is convex, and as we have discussed, $\varphi$ achieves both Fisher (i) and uniform consistency (ii) when $\mathcal{F}$ consists of all measurable functions if and only if $\phi'(0) < 0$ Bartlett et al. (2006).

Now we proceed to consider a restricted hypothesis class, showing in this simple setting that classical consistency fails even when optimal classifiers lie in $\mathcal{F}$, in particular, when $P$ is *optimally predictable* using $\mathcal{F}$, meaning that

$$\text{sgn}(f(x)) = \text{sgn}(\mathbb{P}(Y = 1 \mid X = x) - 1/2). \tag{13}$$

Let $\mathcal{X} = \mathbb{R}^d$ and take $\mathcal{F} = \{f_\theta \mid f_\theta(x) = \langle \theta, x \rangle\}_{\theta \in \mathbb{R}^d}$ to be the collection of linear functionals of $x$. When $P$ is optimally predictable from using $\mathcal{F}$, there exists $\theta^\star$ satisfying $\text{sgn}(\langle \theta^\star, x \rangle) = \text{sgn}(P(Y =$

$1 \mid x) - \frac{1}{2})$, and $f_{\theta^\star}$ minimizes $R(f)$ across all measurable functions. In this case, we say that $P$ is *optimally predictable along $\theta^\star$*. One might expect a margin-based surrogate $\varphi$ achieving Fisher consistency in the classical setup should still consistent. This fails. Even more, for any nonnegative loss $\phi$, there is a data distribution $P$ on $(X, Y)$ such that $\theta_\varphi = \mathrm{argmin}_\theta \mathbb{E}_P[\varphi(f_\theta(X), Y)]$ is essentially orthogonal to $\theta^\star$:

**Proposition 5.** *For any $\epsilon > 0$ and nonzero vector $\theta^\star \in \mathbb{R}^d$, there exists an $(X, Y)$ distribution $P$, optimally predictable along $\theta^\star$, such that for all*

$$\theta_\varphi \in \mathrm{argmin}_\theta R_\varphi(f_\theta) = \mathrm{argmin}_\theta \mathbb{E}\left[\phi(f_\theta(X), Y)\right],$$

*we have $R(f_{\theta_\varphi}) > R(f_{\theta^\star})$ and $|\cos \angle(\theta_\varphi, \theta^\star)| = |\langle \theta_\varphi, \theta^\star \rangle| / (\|\theta_\varphi\|_2 \|\theta^\star\|_2) \le \epsilon$.*

We postpone the proof to Appendix F.1.

Data aggregation methods provide one way to circumvent the the inconsistency Proposition 5 highlights. To state the result, define the approximate minimizers

$$\epsilon\text{-argmin}\, g = \epsilon\text{-argmin}_\theta\, g(\theta) := \left\{\theta \mid g(\theta) \le \inf_\theta g(\theta) + \epsilon\right\}.$$

Suppose the data collection consists of independent samples $Z = (Y_1, \ldots, Y_m)$ and we take $A_m(Z)$ to be majority vote. For a sequence $\epsilon_m$ take

$$\theta_m \in \epsilon_m\text{-argmin}_\theta R_{\varphi, A_m}(f_\theta) = \epsilon_m\text{-argmin}_\theta \mathbb{E}\left[\phi(A_m(Y_1, \ldots, Y_m) f_\theta(X))\right].$$

Then as a corollary to the coming Theorem 3, $f_{\theta_m}$ are asymptotically consistent when $m \to \infty$.

**Corollary B.1.** *Let $P$ be optimally predictable along $\theta^\star$. Then if $\epsilon_m \to 0$ as $m \to \infty$, $R(f_{\theta_m}) \to R(f_{\theta^\star})$ and $\cos \angle(\theta_m, \theta^\star) \to 1$.*

So without aggregation, surrogate risk minimization is (by Proposition 5) essentially arbitrarily incorrect when restricting to the class of linear predictors, while with aggregation, we retain consistency.

## B.2 Aggregation, consistency, and restricted hypothesis classes

As Proposition 5 shows, surrogate risk consistency reposes quite fundamentally on $\mathcal{F}$ containing all measurable functions. We now consider multiclass classification problems, where $\mathcal{Y} = \{1, \ldots, k\}$, and in which $\mathcal{F}$ forms a linear cone satisfying

$$f(x)^T \mathbf{1} = 0 \ \text{ and } \ tf \in \mathcal{F} \text{ for } t > 0 \text{ if } f \in \mathcal{F}.$$

We consider the zero-one loss $\ell(y, y') = 1\{y \ne y'\}$ and $\mathsf{d}(s) = \mathrm{argmax}_{y \in [k]} s_y$, making the restriction to predictors normalized to have $f(x)^T \mathbf{1} = 0$ immaterial. Assume the surrogate $\varphi : \mathbb{R}^k \times [k] \to \mathbb{R}_+$ is Fisher-consistent (i) and satisfies the limiting loss condition

$$\varphi(s, y) \to 0 \ \text{ if } \ s_y - s_j \to +\infty \text{ for all } j \ne y. \tag{14}$$

Many familiar surrogate losses are Fisher consistent and satisfy (14), including the multiclass logistic loss $\varphi(s, y) = \log(\sum_{j=1}^k e^{s_j - s_y})$ and any loss of the form

$$\varphi(s, y) = \sum_{i \ne y} \phi(s_y - s_i)$$

for $\phi$ convex, non-increasing with $\phi'(0) < 0$, and $\inf_t \phi(t) = 0$. Zhang (2004b, Thm. 5) shows that any such loss is consistent over the class $\mathcal{F} = \{f : \mathcal{X} \to \mathbb{R}^k \mid \mathbf{1}^T f = 0\}$. Clearly, the margin-based binary setting in Sec. **??** falls into this scenario when we take $f(x) = (g(x), -g(x))$ for a measurable $g$. Additionally, in a parametric setting when $\mathcal{X} = \mathbb{R}^d$, if $\mathcal{F}$ consists of linear functions $f(x) = (\langle \theta_1, x \rangle, \ldots, \langle \theta_k, x \rangle)$ with $\sum_{i=1}^k \theta_i = 0$, then $\mathcal{F}$ is a (convex) cone.

Extending the definition (13) of optimal predictability in the obvious way, we shall say $\mathcal{F}$ can optimally predict $P$ if there exists $f \in \mathcal{F}$, $f : \mathcal{X} \to \mathbb{R}^k$, for which

$$\mathrm{argmax}_y f_y(x) \in \mathrm{argmax}_y P(Y = y \mid X = x) \ \text{ for all } x.$$

The next theorem shows if $\mathcal{Z} = \mathcal{Y}^m$, we aggregate via majority vote $A_m$, and there is a unique $y^\star(x) = \mathrm{argmax}_y P(Y = y \mid x)$, then surrogate risk minimization is consistent whenever $\mathcal{F}$ can optimally predict $P$.

**Theorem 3.** *Let $\mathcal{F}$ be a cone that optimally predicts $P$, and assume that the minimal excess risk* (8) *satisfies $P(\Delta(X) > 0) = 1$. Let $\epsilon_m \geq 0$ satisfy $\epsilon_m \to 0$. Then for any sequence*

$$f_m \in \epsilon_m\text{-}\underset{f \in \mathcal{F}}{\operatorname{argmin}}\, R_{\varphi, A_m}(f) = \epsilon_m\text{-}\underset{f \in \mathcal{F}}{\operatorname{argmin}}\, \mathbb{E}\left[\varphi(f(X), A_m(Y_1, \ldots, Y_m))\right],$$

*we have $R(f_m) \to R^\star$.*

See Appendix F.2 for a proof.

Theorem 3 shows that for a broad class of surrogate problems with a hypothesis class $\mathcal{F}$ that forms a linear cone, we can achieve consistency asymptotically by aggregation as $m \to \infty$. In contrast, as Proposition 5 shows, even in the "simple" case of binary classification, consistency may fail over subclasses $\mathcal{F}$, even when they include the optimal predictor, and the surrogate can be arbitrarily uninformative.

## C  PROOFS RELATED TO THE RANKING EXAMPLES (SEC. 3.1)

### C.1  PROOF OF PROPOSITION 1

The proof relies on a few notions of variational convergence of functions Rockafellar and Wets (1998), which we review presently. Recall that for a sequence of sets $A_n \subset \mathbb{R}^k$,

$$\limsup_n A_n := \left\{ x \in \mathbb{R}^k \mid \liminf_n \operatorname{dist}(x, A_n) = 0 \right\} = \left\{ x \mid \text{there are } y_n \in A_n \text{ s.t. } y_{n(m)} \to x \right\}$$

and that for a function $g$, we define

$$\epsilon\text{-argmin}\, g = \{s \mid g(s) \leq \inf g + \epsilon\}$$

It will be important for us to discuss convergence of minimizers of convex functions, and to that end, we state the following consequence of the results in Rockafellar and Wets (1998), where $\overline{\mathbb{R}} = \mathbb{R} \cup \{+\infty\}$.

**Lemma C.1.** *Let $g_n : \mathbb{R}^k \to \overline{\mathbb{R}}$ be convex functions with pointwise limit $g$ where $g$ is coercive. Then $g_n$ converges uniformly to $g$ on compacta, $g$ is convex, the $g_n$ are eventually coercive, and for any sequence $\epsilon_n \downarrow 0$ (including those with $\epsilon_n = 0$),*

$$\emptyset \neq \limsup_n \left\{ \epsilon_n\text{-argmin}\, g_n \right\} \subset \operatorname{argmin} g.$$

**Proof**  First, we observe that $g_n \to g$ pointwise implies that $g_n \to g$ uniformly on compacta, and $g$ is convex (see Hiriart-Urruty and Lemaréchal (1993, Thm. IV.3.15) and Rockafellar and Wets (1998, Thm. 7.17)). This is then equivalent to epigraphical convergence of $g_n$ to $g$ (Rockafellar and Wets, 1998, Thm. 7.17). Moreover, as $g_n \to g$ uniformly on compacta, if $x_n \to x$ then $g_n(x_n) \to g(x)$. Thus, for any $\epsilon_n \downarrow 0$, if for a subsequence $n(m) \subset \mathbb{N}$ we have $x_{n(m)} \in \epsilon_{n(m)}\text{-argmin}\, g_n$, and $x_{n(m)} \to x$, we certainly have $g_{n(m)}(x_{n(m)}) \to g(x)$. Consequently (Rockafellar and Wets, 1998, Prop. 7.30) we have

$$\limsup_n \left\{ \epsilon_n\text{-argmin}\, g_n \right\} \subset \operatorname{argmin} g.$$

That the limit supremum is non-empty is then a consequence of (Rockafellar and Wets, 1998, Thm. 7.33), as the convex functions $g_n$ must be coercive as they are convex and $g$ is.  □

We now outline our approach and leverage a few consequences of Lemma C.1. Recall our restriction of $\varphi$ to the set $s^T \mathbf{1} = 0$. For probabilities $p = (p_{ij})_{i,j \leq k}$, define

$$R_\varphi(s \mid p) := \mathbb{E}_{Y \sim p}\left[\varphi(s, Y)\right].$$

We argue that for appropriate $p$, $R_\varphi$ is coercive, and then use Lemma C.1 to argue about the structure of its minimizers. Assume for the sake of contradiction that $\varphi$ is consistent. By considering a distribution $p$ supported only on the pair $(i, j)$, appealing to standard results on surrogate risk consistency for binary decision problems (Bartlett et al., 2006) shows that $\varphi(s, (i, j)) \to \infty$ whenever $(s_j - s_i) \to \infty$.

Now, consider any distribution $p$ containing a cycle over all $i \in \{1, \ldots, k\}$, meaning that there exists a permutation $\pi : [k] \to [k]$ such that $p_{\pi(i),\pi(i+1)} > 0$ for all $i$ (where $\pi(k+1) = \pi(1)$). Then

$$R_\varphi(s \mid p) \geq \min_{i \in [k]} p_{\pi(i),\pi(i+1)} \varphi(s, (\pi(i), \pi(i+1))),$$

and without loss of generality, we assume $\pi(i) = i$. If $\|s\| \to \infty$ while $\mathbf{1}^T s = 0$ (recall the assumption in the proposition), it must be the case that $\max_i(s_{i+1} - s_i) \to \infty$, and so $s \mapsto R_\varphi(s \mid p)$ is coercive whenever $p$ contains a cycle and the minimizers of $R_\varphi(\cdot \mid p)$ exist.

With these preliminaries, we turn to the proposition proper. We construct a distribution $p \in \mathbb{R}_+^{k \times k}$ for which $\mathbf{0}$ must be a minimizer of $R_\varphi(s \mid p)$, and use this to show that $\mathbf{0}$ minimizes $R_\varphi(s \mid (i,j))$ for each pair, yielding a contradiction to Fisher consistency. Consider a distribution $p \in \mathbb{R}_+^{k \times k}$ parameterized by $q \in \mathbb{R}_{++}^k$, i.e., $q > 0$, satisfying $\mathbf{1}^T q = 1$. Then define $p$ to have entries

$$p_{ij} = \begin{cases} q_l, & \text{if } (i,j) = (l, l+1), \\ q_k = 1 - q_1 - \cdots - q_{k-1} > 0, & \text{if } (i,j) = (k, 1), \\ 0, & \text{otherwise.} \end{cases} \tag{15}$$

The corresponding normalized transition matrix $C := C(q_1, \ldots, q_k)$ then takes the form

$$C_{ij} = \begin{cases} 1, & j = i+1 \text{ or } (i,j) = (k, 1), \\ 0, & \text{otherwise.} \end{cases}$$

Evidently $C\mathbf{1} = \mathbf{1}$.

If $\varphi$ is Fisher consistent, we claim that $\mathbf{0}$ must minimize the conditional surrogate risk

$$R_\varphi(s \mid p) = \mathbb{E}_{Y \sim p}[\varphi(s, Y)] = \sum_{l=1}^k q_l \varphi(s, (l, l+1)). \tag{16}$$

To see this, fix an (arbitrary) permutation $\pi$. We tacitly construct a sequence $p^{(n)} \to p$ with $p^{(n)} \in \mathbb{R}_+^{k \times k}$, $p^{(n)^T} \mathbf{1} = \mathbf{1}$, for which the comparison matrix $C^{(n)}$ with non-diagonal entries

$$C_{ij}^{(n)} = \frac{p_{ij}^{(n)}}{\sum_{l \neq i} p_{lj}^{(n)}}$$

satisfies $[C^{(n)}\mathbf{1}]_{\pi(i)} > [C^{(n)}\mathbf{1}]_{\pi(i+1)}$ for each $i$. (To perform this construction, take scalars $v_1 > \cdots > v_k > 0$, and add $\frac{1}{n} v_i$ to each entry of row $\pi(i)$ in $p$, so that if $v_{\pi^{-1}} = (v_{\pi^{-1}(1)}, \ldots, v_{\pi^{-1}(k)})$, then $p^{(n)} = (p + \mathbf{1}v_{\pi^{-1}}^T/n)/\mathbf{1}^T(p + \mathbf{1}v_{\pi^{-1}}^T/n)\mathbf{1}$. Let $n$ be large.)

The presumed Fisher consistency of $\varphi$ means it must be the case that

$$s^{(n)} \in \operatorname*{argmin}_s R_\varphi(s \mid p^{(n)}) \text{ satisfies } s_{\pi(i)}^{(n)} > s_{\pi(i+1)}^{(\pi,n)} \text{ for each } i.$$

Applying Lemma C.1 for each such sequence and permutation $\pi$, we see that the set of minimizers $\operatorname{argmin}_s R_\varphi(s \mid p)$ of the conditional risk (16) must, for each permutation $\pi$, include a vector $s = s(\pi)$ such that

$$s_{\pi(1)} \geq s_{\pi(2)} \geq \cdots \geq s_{\pi(k)}. \tag{17}$$

As $\operatorname{argmin}_s R_\varphi(s \mid p)$ is a closed convex set, we now apply the following

**Lemma C.2.** *Let $S \subset \{s \in \mathbb{R}^k \mid s^T \mathbf{1} = 0\}$ be a convex set containing a vector of the form* (17) *for each permutation $\pi$. Then $\mathbf{0} \in S$.*

**Proof** We proceed by induction on $k \geq 2$. Certainly for $k = 2$, given vectors $u = (s, -s)$ and $v = (-t, t)$ satisfying $s \geq 0$ and $t \geq 0$, we solve

$$\lambda s + (1 - \lambda) t = 0 \text{ or } \lambda = \frac{t}{s+t} \in [0, 1],$$

giving the base of the induction. Now suppose that the lemma holds for dimensions $2, \ldots, k-1$; we wish to show it holds for dimension $k$. Let $I = \{1, \ldots, k-1\}$ be the first $k-1$ indices, and for a

vector $v \in \mathbb{R}^k$ let $v_I = (v_i)_{i \in I}$. Consider any collection $\{v\} \subset S$ covering the permutations (17); take two subsets $\mathcal{V}^1$ and $\mathcal{V}^2$ of these consisting (respectively) of those $v$ such that $v_k \le v_j$ for all $j \le k-1$ and $v_k \ge v_j$ for all $j \le k-1$. Then by the induction, there exist $\overline{v}^i \in \text{Conv}(\mathcal{V}^i)$, $i = 1, 2$ such that

$$\overline{v}^1 = \begin{bmatrix} a\mathbf{1}_{k-1} \\ s \end{bmatrix} \quad \text{and} \quad \overline{v}^2 = \begin{bmatrix} b\mathbf{1}_{k-1} \\ t \end{bmatrix},$$

where $a(k-1) + s = 0$ while $s \le a$ and $b(k-1) + t = 0$ while $t > b$. As $s \le 0$ and $t > 0$, so setting $\lambda = \frac{t}{t-s}$ gives $\lambda \overline{v}^1 + (1-\lambda)\overline{v}^2 = \mathbf{0}$. $\qquad \square$

In particular, we have shown that $\mathbf{0}$ minimizes the surrogate risk (16), and for any vector $q = (q_1, \ldots, q_k) > 0$ defining $p = p(q)$ and $C$ in (15),

$$\inf_s R_\varphi(s \mid p) = R_\varphi(\mathbf{0} \mid p).$$

Notably, the minimizing vector $\mathbf{0}$ is independent of the parameters $q_1, \ldots, q_k$ defining $p$ and $C$ in (15). For $(i, j) \in \mathcal{Y}$, let $D_{ij} = \partial\varphi(\mathbf{0}, (i, j)) \subset \mathbb{R}^k$ be the set of subgradients at $\mathbf{0}$, which is compact convex and non-empty. Then by the first-order optimality condition for subgradients and construction (16) of the conditional surrogate risk, there exist vectors $g_l \in D_{l,l+1}$ satisfying

$$\sum_{l=1}^{k-1} q_l g_l + \left(1 - \sum_{l=1}^{k-1} q_l\right) g_k = 0 \quad \text{and so} \quad g_k = -\frac{\sum_{l=1}^{k-1} q_l g_l}{1 - \sum_{l=1}^{k-1} q_l} \in D_{k,1}.$$

As the $D_{ij}$ are compact convex, by taking $q_k \uparrow 1$ and $(q_1, \ldots, q_{k-1}) \to \mathbf{0}$, we have

$$\|g_k\|_2 \le \frac{\sum_{l=1}^{k-1} q_l}{1 - \sum_{l=1}^{k-1} q_l} \max_{i,j} \sup_{g \in D_{ij}} \|g\|_2 \to 0.$$

As the $D_{ij}$ are closed convex, we evidently have $\mathbf{0} \in D_{k,1}$, while parallel calculation gives $\mathbf{0} \in D_{l,l+1}$ for each $l$. A trivial modification to the construction (15) to apply to cycles other than $(1, 2, \ldots, k, 1)$ then shows that $\mathbf{0}$ minimizes $\varphi(\cdot, (i, j))$ for all pairs $(i, j)$, violating Fisher consistency.

### C.2 Proof of Proposition 2

The proof relies on the fact when $m \ge k$, the event of observing a comparisons $(i, j_i)$ for each $1 \le i \le k$ has nonzero probability. Conditional on this event, we can obtain an unbiased estimate of $C_x \mathbf{1}$. As $\varphi(s, \star) = 0$, it follows that

$$R_{\varphi,A}(s \mid x) = \mathbb{E}\left[\|s - A(Z)\|_2^2 \mathbf{1}\{A(Z) \ne \star\}\right]$$

When $m \ge k$, $\mathbb{P}(A(Z) \ne \star) > 0$, yielding per-$x$ minimizer

$$s^\star = \arg\min R_{\varphi,A}(s \mid x) = \mathbb{E}\left[\left(\frac{m_{i1}}{m_1} + \cdots + \frac{m_{ik}}{m_k}\right)_{i \in [k]} \mid m_1 > 0, \ldots, m_k > 0\right].$$

Conditioned on fixed positive values of $m_1, \cdots, m_k$,

$$(m_{1j}, \cdots, m_{qj}) \sim \text{Multinom}\left(m_j; \frac{p_{1j}}{\sum_{i=1}^k p_{ij}}, \ldots, \frac{p_{kj}}{\sum_{i=1}^k p_{lj}}\right),$$

so $\mathbb{E}[m_{ij}/m_j] = p_{ij}/\sum_{l=1}^k p_{lj} = (C_x)_{ij}$. As $s^\star = C_x \mathbf{1}$ is unique, Fisher consistency follows.

## D Consistency Proofs

### D.1 Proof of Proposition 3

Our only real assumption is that $(s, x) \mapsto \ell(s, P(\cdot \mid x))$ is jointly measurable in $s$ and $x$. Fix a function $f$. Then for any $\epsilon > 0$,

$$R_{\varphi,A}(f) - R_{\varphi,A}^\star = \int_{\mathcal{X}} \delta_{\varphi,A}(f(x), x) dP(x) \ge \int_{\delta_\ell(f(x),x) \ge \epsilon} \psi_A(\epsilon, x) dP(x).$$

Because $\psi_A(\epsilon, x) > 0$ for each $x$, the measure defined by $d\nu(x) = b(x)dP(x)$ is absolutely continuous with respect to $d\mu(x) = \psi_A(\epsilon, x)dP(x)$. That is, there exists $\delta > 0$ such that $\nu(C) \leq \epsilon$ for all $C \subset \mathcal{X}$ satisfying $\nu(C) \leq \delta$. Assume now that $R_{\varphi,A}(f) - R_{\varphi,A}^{\star} \leq \delta$, so that the set $\mathcal{X}_\epsilon := \{x \mid \delta_\ell(f(x), x) \geq \epsilon\}$, which is measurable by the joint measurability assumption, satisfies $\int_{\mathcal{X}_\epsilon} b(x)dP(x) \leq \epsilon$. We find

$$R(f) - R^{\star} = \int_{\delta_\ell(f(x),x) \geq \epsilon} \delta_\ell(f, x)dP(x) + \int_{\delta_\ell(f(x),x) < \epsilon} \delta_\ell(f(x), x)dP(x)$$

$$\leq \int_{\mathcal{X}_\epsilon} b(x)dP(x) + \epsilon \leq 2\epsilon.$$

In particular, we have shown that $R_{\varphi,A}(f) - R_{\varphi,A}^{\star} \leq \delta$ implies $R(f) - R^{\star} \leq 2\epsilon$, which gives the "hard" direction of Fisher consistency. The converse is trivial by considering a single $x$.

To see the comparison inequality, note that by definition of the calibration function,

$$\psi_A(\delta_\ell(f(x), x)) \leq \overline{\psi}_A(\delta_\ell(f(x), x)) \leq \delta_{\varphi,A}(f(x), x)$$

for all $x \in \mathcal{X}$. The result follows by integrating on both sides w.r.t. $P_X$ and applying Jensen's inequality to $\psi_A$.

### D.2 THE EQUIVALENCE OF THE MAMMEN-TSYBAKOV CONDITIONS ($N_\alpha$) AND ($M_\beta$)

We show the analogue of Bartlett et al. (2006, Thm. 3), essentially mimicking their proof, but including it for completeness.

**Lemma D.1.** *Let $\alpha \in [0, 1]$. A distribution $P$ satisfies condition ($N_\alpha$) if and only if it satisfies condition ($M_\beta$) with $\beta = \frac{\alpha}{\alpha - 1}$, where the constant $\mathsf{C_{MT}}$ may differ in the inequalities.*

**Proof** Let condition ($N_\alpha$) hold. We let $c = \mathsf{C_{MT}}$ for shorthand and assume for notational simplicity that $y^\star(x) = \operatorname{argmin}_y \mathbb{E}[\ell(y, Y) \mid X = x]$ is a singleton. Choose a measurable function $f$ such that

$$f(x) = y^\star(x), \text{ if } \Delta(x) > \epsilon \text{ and } \delta_\ell(f(x), x) = \Delta(x) \text{ if } \Delta(x) \leq \epsilon.$$

For all $\alpha \in [0, 1]$, as $\delta_\ell(f(x), x) = 0$ if $\Delta(x) > \epsilon$,

$$\epsilon \mathbb{P}(\Delta(X) \leq \epsilon) \geq \mathbb{E}[\Delta(X)\mathbb{1}\{\Delta(X) \leq \epsilon\}] = \mathbb{E}\left[\delta_\ell(f(X), X)\right] = R(f) - R^{\star}$$

$$\geq \left(\frac{1}{c}\mathbb{P}(\mathsf{d} \circ f \neq \mathsf{d} \circ f^\star)\right)^{\frac{1}{\alpha}} = \left(\frac{1}{c}\mathbb{P}(0 \leq \Delta(X) \leq \epsilon)\right)^{\frac{1}{\alpha}}.$$

Rearranging terms we see for the constant $c' = c^{\frac{1}{\alpha}}$,

$$\mathbb{P}(0 \leq \Delta(X) \leq \epsilon) \leq (c'\epsilon)^{\alpha/(1-\alpha)},$$

so condition ($M_\beta$) holds with $\beta = \frac{\alpha}{1-\alpha}$. (The result is trivial when $\alpha = 1$, as $\mathbb{P}(\Delta(X) \leq \epsilon) = 0$.)

Now assume condition ($M_\beta$) holds for a value $0 \leq \beta < \infty$, that is, $\mathbb{P}(\Delta(X) \leq \epsilon) \leq (c\epsilon)^\beta$ for all $\epsilon > 0$. Recall the definition (8) of $\Delta(x) = \min\{\delta_\ell(s, x) \mid \mathsf{d}(s) \notin y^\star(x)\}$, so that

$$R(f) - R^{\star} = \mathbb{E}\left[\mathbb{1}\{\mathsf{d} \circ f(X) \neq \mathsf{d} \circ f^\star(X)\}\delta_\ell(f(X), X)\right]$$
$$\geq \mathbb{E}\left[\mathbb{1}\{\mathsf{d} \circ f(X) \neq \mathsf{d} \circ f^\star(X)\}\Delta(X)\right],$$

and again by Markov's inequality for any $\epsilon \geq 0$,

$$\mathbb{E}\left[\mathbb{1}\{\mathsf{d} \circ f(X) \neq \mathsf{d} \circ f^\star(X)\}\Delta(X)\right] \geq \epsilon\mathbb{P}(\mathsf{d} \circ f(X) \neq \mathsf{d} \circ f^\star(X), \Delta(X) > \epsilon)$$
$$\geq \epsilon\left(\mathbb{P}(\mathsf{d} \circ f \neq \mathsf{d} \circ f^\star) - \mathbb{P}(\Delta(X) \leq \epsilon)\right) \quad (18)$$
$$\geq \epsilon\mathbb{P}(\mathsf{d} \circ f \neq \mathsf{d} \circ f^\star) - c^\beta\epsilon^{1+\beta},$$

where the last inequality applies condition ($M_\beta$). Maximizing the right hand side, we set

$$\epsilon = \frac{1}{c}\left(\frac{\mathbb{P}(\mathsf{d} \circ f \neq \mathsf{d} \circ f^\star)}{(1+\beta)}\right)^{\frac{1}{\beta}}$$

we obtain

$$R(f) - R^\star \geq \frac{1}{c} \left( \frac{\mathbb{P}(\mathsf{d} \circ f \neq \mathsf{d} \circ f^\star)}{(1+\beta)} \right)^{\frac{1}{\beta}} \cdot \left( \mathbb{P}(\mathsf{d} \circ f \neq \mathsf{d} \circ f^\star) - \frac{\mathbb{P}(\mathsf{d} \circ f \neq \mathsf{d} \circ f^\star)}{(1+\beta)} \right)$$

$$= \frac{\beta}{c(1+\beta)^{\frac{1+\beta}{\beta}}} \cdot \left( \mathbb{P}(\mathsf{d} \circ f \neq \mathsf{d} \circ f^\star) \right)^{\frac{1+\beta}{\beta}} .$$

Set $c' = (c/\beta)^{\frac{\beta}{1+\beta}}(1+\beta)$, and recognize that $\log(1+\beta) - \frac{\beta}{1+\beta} \log \beta \leq \log 2$ (so that $c'$ is indeed a constant), so that condition ($N_\alpha$) holds with $\alpha = \frac{\beta}{1+\beta}$:

$$\mathbb{P}(\mathsf{d} \circ f \neq \mathsf{d} \circ f^\star) \leq c'(R(f) - R^\star)^{\frac{\beta}{1+\beta}} .$$

When $\beta = \infty$, Condition ($M_\beta$) implies $P(\Delta(X) \leq 1/c) = 0$, so taking $\epsilon = 1/c$ in inequality (18)

$$R(f) - R^\star \geq \epsilon \left( \mathbb{P}(\mathsf{d} \circ f \neq \mathsf{d} \circ f^\star) - \mathbb{P}(0 \leq \Delta(X) \leq \epsilon) \right) = \frac{1}{c} \mathbb{P}(\mathsf{d} \circ f \neq \mathsf{d} \circ f^\star).$$

That is, condition ($N_\alpha$) with $\alpha = 1$ holds. $\qquad\square$

### D.3 PROOF OF THEOREM 1

The proof contains two parts. In the first, we provide a lower bound for the calibration function conditioning on $X = x$. We then use the pointwise calibration function to prove a linear comparison inequality on the data space $\mathcal{X}^M := \{x \in \mathcal{X} : \kappa(x) \leq M\}$ of points with low noise condition number, and then conclude the proof via a coarse risk bound on $\mathcal{X} \setminus \mathcal{X}^M$.

**Part 1: lower bounding the pointwise calibration function.**  Before using properties of majority vote $A_m$, we start by assuming a general aggregation method $A : \mathcal{Z} \to \{a_y\}_{y \in \mathcal{Y}}$. To obtain the desired comparison inequality connecting excess surrogate risk and task risk, we recall the pointwise calibration function (5),

$$\overline{\psi}_A(\epsilon, x) := \inf_{s \in \mathbb{R}^d} \left\{ \delta_{\varphi,A}(s,x) : \delta_\ell(s,x) \geq \epsilon \right\}.$$

To lower bound $\overline{\psi}_A(\epsilon, x)$, we need to lower bound $\delta_{\varphi,A}(s,x)$ provided that $\delta_\ell(s,x) \geq \epsilon$. Because $\delta_\ell(s,x) \geq \epsilon > 0$, it must hold that $\mathsf{d}(s) \neq y^\star$, which makes the following general lower bound, which applies for any aggregation method and identifiable loss, useful:

**Lemma D.2.** *Let $\varphi$ be $(\mathsf{C}_{\varphi,1}, \mathsf{C}_{\varphi,2})$-identifiable (Def. 3.1) with parameters $\{a_y\}_{y \in \mathcal{Y}}$ and assume that $\mathsf{d}(s) \neq y^\star$. Then for any aggregation method $A$,*

$$\delta_{\varphi,A}(s,x) \geq \mathsf{C}_{\varphi,1} - (\mathsf{C}_{\varphi,1} + \mathsf{C}_{\varphi,2})\mathbb{P}(A(Z) \neq a_{y^\star}). \tag{19}$$

**Proof**  For the ground truth label $y^\star = y^\star(x)$, $\mathsf{d}(s_{y^\star}) = y^\star$ by Definition 3.1, and

$$R_{\varphi,A}(s_{y^\star} \mid x) \geq \inf_s R_{\varphi,A}(s \mid x).$$

This allows us to bound the excess surrogate risk by

$$\delta_{\varphi,A}(s,x) = R_{\varphi,A}(s \mid x) - \inf_s R_{\varphi,A}(s \mid x) \geq R_{\varphi,A}(s \mid x) - R_{\varphi,A}(s_{y^\star} \mid x)$$

$$= \mathbb{P}(A(Z) = a_{y^\star})\left(\varphi(s, a_{y^\star}) - \varphi(s_{y^\star}, a_{y^\star})\right) + \sum_{j \neq y^\star} \mathbb{P}(A(Z) = a_j)\left(\varphi(s, a_j) - \varphi(s_{y^\star}, a_j)\right).$$

Because by assumption $\mathsf{d}(s) \neq y^\star$, the $(\mathsf{C}_{\varphi,1}, \mathsf{C}_{\varphi,2})$-identifiability of $\varphi$ implies

$$\varphi(s, a_{y^\star}) - \varphi(s_{y^\star}, a_{y^\star}) \geq \inf_{\mathsf{d} \circ s \neq y^\star} \varphi(s, a_{y^\star}) - \varphi(s_{y^\star}, a_{y^\star}) \geq \mathsf{C}_{\varphi,1},$$

$$\varphi(s, a_j) - \varphi(s_{y^\star}, a_j) \geq \inf_s \varphi(s, a_j) - \varphi(s_{y^\star}, a_j) \geq -\mathsf{C}_{\varphi,2},$$

and therefore

$$\delta_{\varphi,A}(s,x) \geq \mathbb{P}(A(Z) = a_{y^\star})\mathsf{C}_{\varphi,1} - (1 - \mathbb{P}(A(Z) = a_{y^\star}))\mathsf{C}_{\varphi,2}$$

$$= \mathsf{C}_{\varphi,1} - (\mathsf{C}_{\varphi,1} + \mathsf{C}_{\varphi,2})\mathbb{P}(A(Z) \neq a_{y^\star}),$$

which is the lower bound (19). □

Equation (19) shows that to lower bound $\delta_{\varphi,A}(s,x)$ when $\mathsf{d}(s) \neq y^\star$, it is sufficient to show that $A(Z) = a_{y^\star}$ with high probability. For the majority vote (7), as the number of labelers $m$ grow, the probability that $\mathbb{P}(A_m(Z) = a_{y^\star}) \to 1$ by standard concentration once we recall the definition (8) of the excess risk $\Delta(x) = \min_{\mathsf{d}(s) \neq y^\star(x)} \delta_\ell(s,x)$.

**Lemma D.3.** *Let* $\mathrm{card}(\mathcal{Y}) = k$. *For all* $s \in \mathbb{R}^d, x \in \mathcal{X}$ *such that* $\delta_\ell(s,x) \geq \epsilon$,

$$\mathbb{P}(A_m(Z) \neq a_{y^\star}) \leq 2k \exp\left(-m\Delta(x)^2/2\right).$$

**Proof** Applying Hoeffding's inequality, simultaneously for each $y \in \mathcal{Y}$,

$$\left| \frac{1}{m} \sum_{l=1}^m \ell(y, Y_l) - \mathbb{E}[\ell(y,Y) \mid X = x] \right| < \frac{\Delta(x)}{2}$$

with probability at least $1 - 2k \exp\left(-m\Delta(x)^2/2\right)$ as $\ell \in [0,1]$. As $\delta_\ell(s,x) = \mathbb{E}[\ell(\mathsf{d}(s),Y) - \ell(y^\star,Y) \mid X = x]$, we have for all $y \neq y^\star$ that

$$\frac{1}{m} \sum_{l=1}^m \ell(y^\star, Y_l) < \frac{1}{m} \sum_{l=1}^m \ell(y, Y_l).$$

Clearly the majority vote method $A_m(Z) = a_{y^\star}$ in this case. □

We can then substitute Lemma D.3 into (19) and obtain a lower bound. To also incorporate the condition $\delta_\ell(s,x) \geq \epsilon$, we recall the noise condition number (9), which guarantees $\Delta(x) \geq \delta_\ell(s,x)/\kappa(x)$ for all $s \in \mathbb{R}^d$. This implies

$$\delta_{\varphi,A_m}(s,x) \geq \mathsf{C}_{\varphi,1} - 2k(\mathsf{C}_{\varphi,1} + \mathsf{C}_{\varphi,2})e^{-\frac{m\Delta(x)^2}{2}} \geq \mathsf{C}_{\varphi,1} - 2k(\mathsf{C}_{\varphi,1} + \mathsf{C}_{\varphi,2})e^{-\frac{m\delta_\ell(s,x)^2}{2\kappa(x)^2}},$$

and thus

$$\overline{\psi}_A(\epsilon, x) \geq \mathsf{C}_{\varphi,1} - 2k(\mathsf{C}_{\varphi,1} + \mathsf{C}_{\varphi,2})e^{-\frac{m\epsilon^2}{2\kappa(x)^2}}.$$

**Part 2: restricting to** $\mathcal{X}^M$. Now it becomes clear why the error function $e_m(t)$ takes the form in Eq. (10), as whenever

$$\epsilon \geq e_m(\kappa(x)) = \sqrt{\frac{2\kappa(x)^2}{m} \log\left(\frac{4k(\mathsf{C}_{\varphi,1} + \mathsf{C}_{\varphi,2})}{\mathsf{C}_{\varphi,1}}\right)},$$

we must have

$$\overline{\psi}_A(\epsilon, x) \geq \mathsf{C}_{\varphi,1} - 2k(\mathsf{C}_{\varphi,1} + \mathsf{C}_{\varphi,2})e^{-\log\left(\frac{4k(\mathsf{C}_{\varphi,1} + \mathsf{C}_{\varphi,2})}{\mathsf{C}_{\varphi,1}}\right)} \geq \mathsf{C}_{\varphi,1}/2,$$

which further implies a pointwise convex lower bound $\overline{\psi}_A(\epsilon, x) \geq \mathsf{C}_{\varphi,1}(\epsilon - e_m(\kappa(x)))_+/2$. Restricting to $x \in \mathcal{X}^M = \{x \in \mathcal{X} \mid \kappa(x) \leq M\}$, we clearly have

$$\psi_{A_m}^M(\epsilon) := \frac{1}{2}\mathsf{C}_{\varphi,1}[\epsilon - e_m(M)]_+ \leq \overline{\psi}_A(\epsilon, x).$$

Now, we proceed with an argument similar to those Bartlett et al. (2006) use to provide fast rates of convergence in binary classification using $\psi_{A_m}^M(\epsilon)$ and applying over a restricted data space $\mathcal{X}^M$.

**Lemma D.4.** *Let* $M > 0$ *and for* $f \in \mathcal{F}$, *define* $D(f,M) := R(f) - R^\star - \mathbb{P}(\kappa(X) > M)$. *Whenever* $D(f,M) \geq 0$,

$$\mathsf{C}_{\mathsf{MT}}D(f,M)^\alpha \cdot \psi_{A_m}^M\left(\frac{D(f,M)^{1-\alpha}}{2\mathsf{C}_{\mathsf{MT}}}\right) \leq R_{\varphi,A_m}(f) - R_{\varphi,A_m}^\star.$$

**Proof** We begin with some generalities. By condition ($N_\alpha$), for any function $f$ and $\epsilon > 0$,

$$\mathbb{E}[1\{\mathsf{d} \circ f(X) \neq \mathsf{d} \circ f^\star(X), \delta_\ell(f(X), X) < \epsilon\} \delta_\ell(f(X), X)] \leq \mathsf{C}_{\mathsf{MT}}\epsilon \cdot (R(f) - R^\star)^\alpha,$$

so that

$$R(f) - R^\star = \mathbb{E}\left[1\{\mathsf{d} \circ f(X) \neq y^\star(X)\} \delta_\ell(f(X), X)\right]$$
$$\leq \mathsf{C}_{\mathsf{MT}}\epsilon \cdot (R(f) - R^\star)^\alpha + \mathbb{E}\left[1\{\delta_\ell(f(X), X) \geq \epsilon\} \delta_\ell(f(X), X)\right]. \qquad (20)$$

Consider the second term in the bound (20). For any convex function $0 \leq \psi$ with $\psi(0) = 0$, $\epsilon \mapsto \psi(\epsilon)/\epsilon$ is non-decreasing on $\epsilon > 0$ (cf. (Hiriart-Urruty and Lemaréchal, 1993, Ch. 1)). This implies

$$\frac{\psi(\epsilon)}{\epsilon} 1\{\delta_\ell(f(x), x) \geq \epsilon\} \leq \frac{\psi(\delta_\ell(f(x), x))}{\delta_\ell(f(x), x)},$$

where we take $0/0 = 0$, and so $\psi(\epsilon)\delta_\ell(f(x), x)1\{\delta_\ell(f(x), x) \geq \epsilon\} \leq \epsilon \cdot \psi(\delta_\ell(f(x), x))$. Leveraging the calibration function (5), if $\psi(\epsilon) \leq \overline{\psi}_{\varphi, A_m}(\epsilon)$, then we evidently have

$$\psi(\epsilon)\mathbb{E}\left[1\{\delta_\ell(f(X), X) \geq \epsilon\} \cdot \delta_\ell(f(X), X)\right]$$
$$\leq \epsilon \cdot \mathbb{E}\left[\psi(\delta_\ell(f(X), X))\right] \leq \epsilon \cdot \mathbb{E}\left[\delta_{\varphi, A_m}(f(X), X)\right] = \epsilon \left(R_{\varphi, A_m}(f) - R^\star_{\varphi, A_m}\right). \qquad (21)$$

With these generalities in place, consider the function $f^M(x) = f(x)1\{x \in \mathcal{X}^M\} + f^\star(x)1\{x \notin \mathcal{X}^M\}$. Substituting this in inequality (20) yields

$$R(f^M) - R^\star \leq \mathsf{C}_{\mathsf{MT}}\epsilon \cdot (R(f^M) - R^\star)^\alpha + \mathbb{E}[1\{\delta_\ell(f^M(X), X) \geq \epsilon\} \delta_\ell(f^M(X), X)]. \qquad (22)$$

for all $\epsilon > 0$. Because the truncated calibration function $\psi_{A_m}^M$ is convex, inequality (21) yields

$$\psi_{A_m}^M(\epsilon)\mathbb{E}[1\{\delta_\ell(f^M(X), X) \geq \epsilon\} \cdot \delta_\ell(f^M(X), X)] \leq \epsilon \cdot \mathbb{E}[\psi_{A_m}^M(\delta_\ell(f^M(X), X))].$$

Because $0 \leq \psi_{A_m}^M \leq \psi_{A_m} = \overline{\psi}_{A_m}^{**}$, we evidently obtain

$$\mathbb{E}\left[\psi_{A_m}^M(\delta_\ell(f^M(X), X))\right] \leq R_{\varphi, A_m}(f) - R^\star_{\varphi, A_m}.$$

By inequality (22), we therefore have

$$\frac{R(f^M) - R^\star}{\epsilon} - \mathsf{C}_{\mathsf{MT}}\left(R(f^M) - R^\star\right)^\alpha \leq \frac{1}{\epsilon}\mathbb{E}\left[1\{\delta_\ell(f^M(X), X) \geq \epsilon\} \delta_\ell(f^M(X), X)\right],$$

and so multiplying by $\psi_{A_m}^M(\epsilon)$,

$$\psi_{A_m}^M(\epsilon)\left(\frac{R(f^M) - R^\star}{\epsilon} - \mathsf{C}_{\mathsf{MT}}(R(f^M) - R^\star)^\alpha\right)$$
$$\leq \frac{\psi_{A_m}^M(\epsilon)}{\epsilon}\mathbb{E}\left[1\{\delta_\ell(f^M(X), X) \geq \epsilon\} \cdot \delta_\ell(f^M(X), X)\right] \leq R_{\varphi, A_m}(f^M) - R^\star_{\varphi, A_m}. \qquad (23)$$

Finally, we use that $\epsilon$ was arbitrary. Taking $\epsilon = (R(f^M) - R^\star)^{1-\alpha}/(2\mathsf{C}_{\mathsf{MT}})$ in inequality (23) gives $\psi_{A_m}^M(\epsilon)\mathsf{C}_{\mathsf{MT}}(R(f^M) - R^\star)^\alpha \leq R_{\varphi, A_m}(f^M) - R^\star_{\varphi, A_m}$. Using that

$$D(f, M) := R(f) - R^\star - \mathbb{P}(\kappa(X) > M) \leq R(f^M) - R^\star$$

completes the proof of the lemma. $\qquad \square$

We have nearly completed the proof of Theorem 1. By the condition $R(f) - R^\star \geq 2\mathbb{P}(\kappa(X) > M) + (4\mathsf{C}_{\mathsf{MT}}e_m(M))^{\frac{1}{1-\alpha}}$ we have $D(f, M) \geq (R(f) - R^\star)/2$, while at the same time

$$\frac{D(f, M)^{1-\alpha}}{2\mathsf{C}_{\mathsf{MT}}} \geq 2e_m(M).$$

By convexity, $\psi_{A_m}^M(\epsilon)/\epsilon$ is non-decreasing in $\epsilon$, so we further have

$$\mathsf{C}_{\mathsf{MT}}D(f, M)^\alpha \psi_{A_m}^M\left(\frac{D(f, M)^{1-\alpha}}{2\mathsf{C}_{\mathsf{MT}}}\right) \geq \mathsf{C}_{\mathsf{MT}}D(f, M)^\alpha \cdot \frac{D(f, M)^{1-\alpha}}{2\mathsf{C}_{\mathsf{MT}}} \cdot \frac{\psi_{A_m}^M(2e_m(M))}{2e_m(M)}$$
$$= \frac{1}{2}D(f, M) \cdot \frac{1}{4}\mathsf{C}_{\varphi, 1} \geq \frac{\mathsf{C}_{\varphi, 1}(R(f) - R^\star)}{16}.$$

Substitute the above display into Lemma D.4.

### D.4 PROOF OF COROLLARY 3.3

Recall that $k = \text{card}(\mathcal{Y}) < \infty$. For the binary case that $k = 2$, we simply take $M = 1$ and as

$$4\mathsf{C}_{\mathsf{MT}} e_m(1) = \left(\frac{32\mathsf{C}_{\mathsf{MT}}^2}{m} \log\left(\frac{8(\mathsf{C}_{\varphi,1} + \mathsf{C}_{\varphi,2})}{\mathsf{C}_{\varphi,1}}\right)\right)^{\frac{1}{2(1-\alpha)}} = \xi_{m,2},$$

Theorem 1 implies the conclusion.

For the general multiclass case, we can bound the tail probability by using $\kappa(X) \leq 1/\Delta(X)$ and the low noise condition (N$_\alpha$), as $\mathbb{P}(\kappa(X) > M) \leq \mathbb{P}(\Delta(X) \leq 1/M) \leq (\mathsf{C}_{\mathsf{MT}}/M)^{\frac{\alpha}{1-\alpha}}$. Therefore, using Theorem 1, we only need to prove

$$\inf_M \{2 \cdot (\mathsf{C}_{\mathsf{MT}}/M)^{\frac{\alpha}{1-\alpha}} + (4\mathsf{C}_{\mathsf{MT}} e_m(M))^{\frac{1}{1-\alpha}}\} \leq 4 \cdot \left(\frac{32\mathsf{C}_{\mathsf{MT}}^4}{m} \log\left(\frac{4k(\mathsf{C}_{\varphi,1} + \mathsf{C}_{\varphi,2})}{\mathsf{C}_{\varphi,1}}\right)\right)^{\frac{\alpha}{2(1-\alpha^2)}}.$$

Indeed, we choose the $M$ such that $2(\mathsf{C}_{\mathsf{MT}}/M)^{\frac{\alpha}{1-\alpha}} = (4\mathsf{C}_{\mathsf{MT}} e_m(M))^{\frac{1}{1-\alpha}}$, which, by substituting in Eq. (10), is equivalent to

$$M^{-\frac{1+\alpha}{1-\alpha}} = \left(\frac{32}{m} \log\left(\frac{4k(\mathsf{C}_{\varphi,1} + \mathsf{C}_{\varphi,2})}{\mathsf{C}_{\varphi,1}}\right)\right)^{\frac{1}{2(1-\alpha)}} \cdot \frac{\mathsf{C}_{\mathsf{MT}}}{2},$$

and thus we choose

$$M = \left(\frac{32}{m} \log\left(\frac{4k(\mathsf{C}_{\varphi,1} + \mathsf{C}_{\varphi,2})}{\mathsf{C}_{\varphi,1}}\right)\right)^{-\frac{1}{2(1+\alpha)}} \cdot \left(\frac{\mathsf{C}_{\mathsf{MT}}}{2}\right)^{-\frac{1-\alpha}{1+\alpha}}.$$

With this choice

$$2 \cdot (\mathsf{C}_{\mathsf{MT}}/M)^{\frac{\alpha}{1-\alpha}} + (4\mathsf{C}_{\mathsf{MT}} e_m(M))^{\frac{1}{1-\alpha}}$$

$$= 2 \cdot (4\mathsf{C}_{\mathsf{MT}} e_m(M))^{\frac{1}{1-\alpha}}$$

$$= 2 \cdot \left(\frac{32\mathsf{C}_{\mathsf{MT}}^2}{m} \log\left(\frac{4k(\mathsf{C}_{\varphi,1} + \mathsf{C}_{\varphi,2})}{\mathsf{C}_{\varphi,1}}\right)\right)^{\frac{1}{2(1-\alpha)}} \cdot \left(\frac{32}{m} \log\left(\frac{4k(\mathsf{C}_{\varphi,1} + \mathsf{C}_{\varphi,2})}{\mathsf{C}_{\varphi,1}}\right)\right)^{-\frac{1}{2(1+\alpha)(1-\alpha)}} \cdot \left(\frac{\mathsf{C}_{\mathsf{MT}}}{2}\right)^{-\frac{1}{1+\alpha}}$$

$$= 2^{1+\frac{1}{1+\alpha}} \cdot \left(\frac{32}{m} \log\left(\frac{4k(\mathsf{C}_{\varphi,1} + \mathsf{C}_{\varphi,2})}{\mathsf{C}_{\varphi,1}}\right)\right)^{\frac{1}{2(1-\alpha)}} \cdot \left(\frac{32}{m} \log\left(\frac{4k(\mathsf{C}_{\varphi,1} + \mathsf{C}_{\varphi,2})}{\mathsf{C}_{\varphi,1}}\right)\right)^{-\frac{1}{2(1+\alpha)(1-\alpha)}} \cdot \mathsf{C}_{\mathsf{MT}}^{\frac{1}{1-\alpha} - \frac{1}{1+\alpha}}$$

$$\leq 4 \cdot \left(\frac{32}{m} \log\left(\frac{4k(\mathsf{C}_{\varphi,1} + \mathsf{C}_{\varphi,2})}{\mathsf{C}_{\varphi,1}}\right)\right)^{\frac{\alpha}{2(1-\alpha^2)}} \cdot \mathsf{C}_{\mathsf{MT}}^{\frac{2\alpha}{1-\alpha^2}}$$

$$= 4 \cdot \left(\frac{32\mathsf{C}_{\mathsf{MT}}^4}{m} \log\left(\frac{4k(\mathsf{C}_{\varphi,1} + \mathsf{C}_{\varphi,2})}{\mathsf{C}_{\varphi,1}}\right)\right)^{\frac{\alpha}{2(1-\alpha^2)}} = \xi_{m,k}.$$

### D.5 PROOFS FOR THE IDENTIFIABLE SURROGATE LOSSES

#### D.5.1 PROOF OF LEMMA A.1

That $\phi(\delta) < 0$ is immediate because $\phi$ is non-increasing by assumption, and the monotonicity properties of convex functions (Hiriart-Urruty and Lemaréchal, 1993, Ch. 1) guarantee it strictly decreases near 0. For $y \in \{\pm 1\}$, we take $s_y = \delta y$ and $a_y = y$, and

$$\varphi(s_y, a_y) + \mathsf{C}_{\varphi,1} = \phi(\delta) + \mathsf{C}_{\varphi,1} = \phi(0) = \inf_{\mathsf{d}(s) \neq y} \varphi(s, a_y)$$

$$\varphi(s_y, a_{-y}) - \mathsf{C}_{\varphi,2} \leq \phi(-\delta) - \mathsf{C}_{\varphi,2} = 0 = \inf_{s \in \mathbb{R}} \varphi(s, a_y).$$

by direct evaluation.

### D.5.2 PROOF OF LEMMA A.2

For each $y \in \mathcal{Y}$, we choose $(s_y, a_y) = (v(y)/N, y)$. Observe that for each graph $y \in \mathcal{Y}$,

$$\varphi(s_y, a_y) = \max_{\hat{y} \in \mathcal{Y}} \left( \frac{1}{2N} \|v(\hat{y}) - v(y)\|_2^2 + \langle v(\hat{y}) - v(y), v(y)/N \rangle \right)$$

$$= \max_{\hat{y} \in \mathcal{Y}} \left( \frac{1}{2N} \|v(\hat{y})\|_2^2 - \frac{1}{2N} \|v(y)\|_2^2 \right)$$

$$= 0,$$

where we use $\|v(y)\|_2^2 = N$ for all $y \in \mathcal{Y}$. If $\mathsf{d}(s) \neq y$, then there exists some $y' \neq y \in \mathcal{Y}$ such that

$$\langle v(y') - v(y), s \rangle \geq 0.$$

This implies

$$\varphi(s, a_y) = \varphi(s, y) \geq \left( \frac{1}{2N} \|v(y') - v(y)\|_1 + \langle v(y') - v(y), s \rangle \right) \geq \frac{1}{2N} \|v(y') - v(y)\|_1 \geq \frac{1}{N}$$

because distinct bipartite matchings differ on at least two edges. This then implies condition (6a) holds with $\mathsf{C}_{\varphi,1} = 1/N$. The second condition (6b) holds for $\mathsf{C}_{\varphi,2} = 2$ as

$$\varphi(s_{y'}, a_y) = \varphi(v(y')/N, y) = \max_{\hat{y} \in \mathcal{Y}} \left( \frac{1}{2N} \|v(\hat{y}) - v(y)\|_1 + \langle v(\hat{y}) - v(y), v(y')/N \rangle \right) \leq 2$$

whenever $v^T \mathbf{1} = N$.

### D.5.3 PROOF OF LEMMA A.3

By definition of $\tau(y)$, for any $\epsilon > 0$ and each $y \in \mathcal{Y}$, we can take $s_y \in \mathcal{S}(y)$ (by using homogeneity and scaling) such that

$$\max_{\hat{y} \neq y} \ell(\hat{y}, y) / \langle v(y) - v(\hat{y}), s_y \rangle = 1 \text{ and } \min_{\hat{y} \neq y} \ell(\hat{y}, y) / \langle v(y) - v(\hat{y}), s_y \rangle > \frac{1}{\tau(y) + \epsilon}. \tag{24}$$

We take $a_y = y$.

**Controlling $\mathsf{C}_{\varphi,1}$.** Because by assumption $\max_{\hat{y} \neq y} \ell(\hat{y}, y) / \langle v(y) - v(\hat{y}), s \rangle = 1$,

$$\varphi(s_y, y) = \max_{\hat{y} \in \mathcal{Y}} \left( \ell(\hat{y}, y) + \langle v(\hat{y}) - v(y), s_y \rangle \right)_+$$

$$= \max_{\hat{y} \in \mathcal{Y}} \left\{ \langle v(y) - v(\hat{y}), s_y \rangle \cdot \left( \ell(\hat{y}, y) / \langle v(y) - v(\hat{y}), s_y \rangle - 1 \right)_+ \right\} = 0.$$

For any $s$ such that $\mathsf{d}(s) \neq y$, there must exist $\hat{y} \neq y$ such that $\langle v(y) - v(\hat{y}), s \rangle \geq 0$ and thus

$$\varphi(s, y) \geq \ell(\hat{y}, y) + \langle v(y) - v(\hat{y}), s \rangle \geq \min_{\hat{y} \neq y} \ell(\hat{y}, y) = \min_{\hat{y} \neq y} \ell(\hat{y}, y) + \varphi(s_y, y).$$

Thus we can take $\mathsf{C}_{\varphi,1} = \min_{\hat{y} \neq y} \ell(\hat{y}, y)$.

**Controlling $\mathsf{C}_{\varphi,2}$.** For any $y' \neq y$, the $s_y$ satisfying inequality (24) yields

$$\langle v(\hat{y}) - v(y), s_{y'} \rangle = \langle v(\hat{y}) - v(y'), s_{y'} \rangle + \langle v(y') - v(y), s_{y'} \rangle \leq \langle v(y') - v(y), s_{y'} \rangle - \ell(\hat{y}, y')$$

$$\leq (\tau(y') + \epsilon) \cdot \ell(y, y') - \ell(\hat{y}, y'),$$

By the normalization $0 \leq \ell \leq 1$, we have

$$\varphi(s_{y'}, a_y) = \max_{\hat{y} \in \mathcal{Y}} \left( \ell(\hat{y}, y) + \langle v(\hat{y}) - v(y), s_{y'} \rangle \right)_+$$

$$\leq \max_{\hat{y} \in \mathcal{Y}} \left( \ell(\hat{y}, y) + (\tau(y') + \epsilon) \cdot \ell(y, y') \right)_+ \leq \tau(y') + 1 + \epsilon.$$

As $\epsilon > 0$ was arbitrary, we can take $\mathsf{C}_{\varphi,2} = \max_{y \in \mathcal{Y}} \tau(y) + 1$.

**The special case of $\ell_0$ task loss.** Finally we are left to show if $v(y) \in \{0,1\}^d$ and $\ell(\hat{y}, y) = \frac{1}{2d} \|\hat{y} - y\|_1$, we have $\tau(y) = 1$ for all $y$. This is trivial in this case, as we can take $s_y = 2v(y) - 1$, and for $\hat{y} \neq y$,

$$\langle v(y) - v(\hat{y}), s_y \rangle = \langle v(y) - v(\hat{y}), 2v(y) - 1 \rangle = \langle v(y) - v(\hat{y}), 2v(y) \rangle - \|v(y)\|_2^2 + \|v(\hat{y})\|_2^2$$

$$= \|v(y)\|_2^2 - 2\langle v(y), v(\hat{y}) \rangle + \|v(\hat{y})\|_2^2$$

$$= \|v(y) - v(\hat{y})\|_2^2 = \|v(y) - v(\hat{y})\|_0 = 2d\ell(\hat{y}, y),$$

where we again use the fact that for 0-1 features, $\|v(y) - v(\hat{y})\|_2^2 = \|v(y) - v(\hat{y})\|_1$. We see that $\ell(\hat{y}, y)/\langle v(y) - v(\hat{y}), s_y \rangle$ is a constant and thus $\tau(y) = 1$ for all $y \in \mathcal{Y}$.

## E $K$-NEAREST-NEIGHBORS AND GENERAL AGGREGATION METHODS

In this section, we adapt the results in Section 3.2 to demonstrate a consistency result for $K$-nearest-neighbor methods using an analogue of majority vote labeling. We assume the surrogate $\varphi$ is $(\mathsf{C}_{\varphi,1}, \mathsf{C}_{\varphi,2})$-identifiable (Def. 3.1) with parameters $\{a_y\}_{y \in \mathcal{Y}}$ and that $k = \mathrm{card}(\mathcal{Y}) < \infty$. Given a sample $(X_i, Y_i)_{i=1}^n$ and a point $x \in \mathcal{X}$, sort the indices so that $\mathrm{dist}(X_{(1)}, x) \leq \mathrm{dist}(X_{(2)}, x) \leq \cdots \mathrm{dist}(X_{(n)}, x)$ (and label $Y_{(i)}$) similarly. Then the nearest-neighbor aggregator of a point $x$ is

$$A_{n,K}(x) := a_{\hat{y}}, \ \hat{y} = \underset{y \in \mathcal{Y}}{\mathrm{argmin}} \sum_{i=1}^K \ell(y, Y_{(i)}), \tag{25}$$

and we define the surrogate risk

$$R_{\varphi,n,K}(f) := \mathbb{E}[\varphi(f(X), A_{n,K}(X))],$$

where $A_{n,K}$ implicitly depends on an imagined sample $(X_i, Y_i)_{i=1}^n$. We warn the reader that, at some level, the surrogate consistency guarantee we provide will implicitly essentially show that $K$-nearest-neighbors is consistent so long as $K \to \infty$ while $K/n \to 0$, a familiar result for multiclass classification and regression problems Stone (1977); Devroye et al. (1996).

We will demonstrate the following theorem.

**Theorem 4.** *Let the loss $\varphi$ be identifiable (Definition 3.1), assume the excess risk (8) satisfies $P(\Delta(X) > 0) = 1$. Let $K = K(n)$ and $n$ satisfy $K/n \to 0$ and $K \to \infty$ as $n \to \infty$. Then for all $\epsilon > 0$, there exists $N$ and $\delta > 0$ such that for all $n \geq N$,*

$$R_{\varphi,n,K}(f) - R_{\varphi,n,K}^\star \leq \delta \ \text{implies} \ R(f) - R^\star \leq \epsilon$$

*for all measurable $f$.*

The theorem more or less follows from the following comparison inequality.

**Lemma E.1.** *Let $\varphi$ be $(\mathsf{C}_{\varphi,1}, \mathsf{C}_{\varphi,2})$-identifiable, $\gamma > 0$ satisfy $\gamma \leq \frac{\mathsf{C}_{\varphi,1}}{2(\mathsf{C}_{\varphi,1} + \mathsf{C}_{\varphi,2})}$, and define the set*

$$\mathcal{X}^\gamma_{n,K} := \{x \in \mathcal{X} \mid \mathbb{P}(A_{n,K}(x) \neq a_{y^\star(x)}) \leq \gamma\}.$$

*Then for all measurable $f$,*

$$R(f) - R^\star \leq \frac{2}{\mathsf{C}_{\varphi,1}} \left( R_{\varphi,n,K}(f) - R_{\varphi,n,K}^\star(f) \right) + \mathbb{P}(X \notin \mathcal{X}^\gamma_{n,K}).$$

**Proof** For $n, K \in \mathbb{N}$, define the pointwise risk gap

$$\delta_{\varphi,n,K}(s, x) := \mathbb{E}\left[\varphi(s, A_{n,K}(x))\right] - \inf_{s'} \mathbb{E}\left[\varphi(s, A_{n,K}(x))\right],$$

where the expectation is over the nearest-neighbor aggregation (25), and for $\epsilon > 0$ define the pointwise calibration function

$$\psi_{n,K}(\epsilon, x) := \inf_{s \in \mathbb{R}^d} \left\{\delta_{\varphi,n,K}(s, x) \mid \delta_\ell(s, x) \geq \epsilon\right\}.$$

Because Lemma D.2 (in the proof of Theorem 1) holds for any aggregation method, we see that

$$\psi_{n,K}(\epsilon, x) \geq \mathsf{C}_{\varphi,1} - (\mathsf{C}_{\varphi,1} + \mathsf{C}_{\varphi,2})\mathbb{P}(A_{n,K}(x) \neq a_{y^\star(x)})$$

for all $x \in \mathcal{X}$ and $\epsilon > 0$. Let $\mathcal{X}^\gamma = \mathcal{X}_{n,K}^\gamma$ for shorthand. Then in particular, because $\gamma > 0$ is small enough that $(\mathsf{C}_{\varphi,1} + \mathsf{C}_{\varphi,2})\gamma < \mathsf{C}_{\varphi,1}/2$, we have $\psi_{n,K}(\epsilon, x) \geq \frac{1}{2}\mathsf{C}_{\varphi,1}$ for $x \in \mathcal{X}^\gamma$. As a consequence, we can expand the risk

$$
\begin{aligned}
R(f) - R^\star &= \mathbb{E}[\delta_\ell(f(X), X)] \\
&\leq \mathbb{E}\left[\delta_\ell(f(X), X)1\{\delta_\ell(f(X), X) \geq \epsilon\}\right] + \epsilon \\
&\leq \mathbb{E}\left[\delta_\ell(f(X), X)1\{\delta_\ell(f(X), X) \geq \epsilon, X \in \mathcal{X}^\gamma\}\right] + \mathbb{P}(X \notin \mathcal{X}^\gamma) + \epsilon.
\end{aligned}
$$

As we assume $\ell \leq 1$, we see that $\psi_{n,K}(\delta_\ell(f(x), x), x) \geq \frac{1}{2}\mathsf{C}_{\varphi,1} \cdot \delta_\ell(f(x), x)$ when $\delta_\ell(f(x), x) \geq \epsilon$ and $x \in \mathcal{X}^\gamma$, giving the upper bound

$$
\begin{aligned}
R(f) - R^\star &\leq \frac{2}{\mathsf{C}_{\varphi,1}}\mathbb{E}\left[\psi_{n,K}(\delta_\ell(f(X), X))1\{X \in \mathcal{X}^\gamma\}\right] + \mathbb{P}(X \notin \mathcal{X}^\gamma) + \epsilon \\
&\leq \frac{2}{\mathsf{C}_{\varphi,1}}\left(R_{\varphi,n,K}(f) - R_{\varphi,n,K}^\star(f)\right) + \mathbb{P}(X \notin \mathcal{X}^\gamma) + \epsilon.
\end{aligned}
$$

As $\epsilon > 0$ was arbitrary we obtain the lemma. $\qquad\square$

By Lemma E.1, it is therefore sufficient to show that for any fixed $\gamma > 0$, $\mathbb{P}(X \notin \mathcal{X}_{n,K}^\gamma) \to 0$. But for this, we can simply rely on the results of Stone (1977): by his Theorems 1 and 2, because $\mathcal{Y}$ is finite, $K$-nearest neighbors (when $K = K(n)$ satisfies $K \to \infty$ and $K/n \to 0$) is consistent for estimating the conditional distribution of $P(Y = y \mid X = x)$. Because $\Delta(X) > 0$ with probability 1, we see that $\mathbb{P}(A_{n,K}(x) \neq a_{y^\star(x)}) \to 0$ for all $x$ except perhaps a null set, and so Stone's results imply $\mathbb{P}(X \notin \mathcal{X}_{n,K}^\gamma) \to 0$.

# F  PROOFS ASSOCIATED WITH MODEL-BASED CONSISTENCY

## F.1  PROOF OF PROPOSITION 5

We begin by considering the distribution $P_{x_1,x_2}$, whose $X$-marginal is supported only on two data points $\{x_1, x_2\} \subset \mathcal{X}$. The key idea is that by carefully choosing $x_1, x_2$ and the conditional distribution of $Y \mid X = x$, the conditional surrogate losses

$$
R_\varphi(t \mid x_i) := \mathbb{E}[\varphi(f_{t\theta^\star}(X), Y) \mid X = x_i] = \mathbb{E}[\phi(Y\langle t\theta^\star, X\rangle) \mid X = x_i], \quad i = 1, 2,
$$

attain their minima at distinct $t$, and if their is a $\theta \notin \text{span}\{\theta^\star\}$ for which $\mathbb{E}[\varphi(f_\theta(X), Y) \mid X = x_i] = \inf_t \mathbb{E}[\phi(Yt) \mid X = x_i]$ for each $i$, then $f_\theta$ would attain less surrogate risk than any point in $\text{span}\{\theta^\star\}$. To guarantee that $R(\theta) = P(Y\langle X, \theta\rangle \leq 0)$ has a unique up to scaling—that only points in $\text{span}\{\theta^\star\}$ minimize $R$—we perturb $P_{x_1,x_2}$ slightly by defining $P$ to have $X$-marginal

$$
P(X \in \cdot) = \frac{1-\delta}{2}\left[\boldsymbol{\delta}_{x_1} + \boldsymbol{\delta}_{x_2}\right] + \delta\mathsf{N}(0, I_d),
$$

where $\boldsymbol{\delta}_x$ denotes a point mass at $x$ and $\delta \geq 0$ is a value to be chosen.

**The construction of $P_{x_1,x_2}$ and $P$.**  Without loss of generality, we take $\theta^\star = e_1$, the first canonical basis vector. For a value $\beta > 0$ to be defined, define the $Y$ conditional probability

$$
\eta_\beta(x) = P(Y = 1 \mid X = x) := \min\left\{\left[\frac{1}{2} + \langle x, e_1\rangle(\beta|\langle x, e_2\rangle| + 1)\right]_+, 1\right\}
$$

which projects $\frac{1}{2} + \langle x, e_1\rangle(\beta|\langle x, e_2\rangle| + 1)$ onto $[0, 1]$ and satisfies $\eta_\beta(x) < \frac{1}{2}$ if and only if $\langle x, e_1\rangle < 0$ and $\eta_\beta(x) > \frac{1}{2}$ if and only if $\langle x, e_1\rangle > 0$. With this construction, $\theta^\star = e_1$ is evidently the unique unit vector $u \in \mathbb{S}^{d-1}$ satisfying $\text{sgn}(\langle x, u\rangle) = \text{sgn}(\eta_\beta(x) - 1/2)$ for all $x$, so for any $\theta \notin \text{span}\{\theta^\star\}$,

$$
R(f_\theta) > R(f_{\theta^\star}).
$$

We can now provide the explicit construction of the distribution $P$. Assume we may take the two points $x_1, x_2$ to satisfy $\eta_\beta(x_1) = \frac{2}{3}$ and $\eta_\beta(x_2) = \frac{1}{3}$. Then defining

$$
g_\phi(t) = \frac{2}{3}\phi(t) + \frac{1}{3}\phi(-t),
$$

which is a coercive convex function (and so has a compact interval of minimizers), we write the surrogate risk of a vector $\theta$ for $P_{x_1,x_2}$ (recalling that $P(Y = y \mid x) = \eta_\beta(x)$) as

$$\mathbb{E}_{P_{x_1,x_2}}[\varphi(f_\theta(X), Y)] = \frac{1}{2}g_\phi(\langle\theta, x_1\rangle) + \frac{1}{2}g_\phi(-\langle\theta, x_2\rangle).$$

By direct calculation, for any $\alpha > \frac{1}{2}$ and $\beta > 0$, the choices

$$x_1 = \frac{1}{6}e_1 \text{ and } x_2 = -\frac{1}{12\alpha}e_1 + \frac{2\alpha - 1}{\beta}e_2 \tag{26}$$

guarantee $\eta_\beta(x_1) = \frac{2}{3}$ and $\eta_\beta(x_2) = \frac{1}{3}$.

**Minimizing surrogate risk along certain direction.** We wish to show that the surrogate attains its minimum along a direction $u$ nearly perpendicular to $\mathrm{span}\{\theta^\star\}$. Let $u \in \mathbb{S}^{d-1}$ have coordinates $u_j = \langle u, e_j\rangle$. We shall prove the following lemma:

**Lemma F.1.** *Let $x_1, x_2$ have definition (26), $\beta > 0$, and $P$ be defined as above. Then $\theta^\star = e_1$ yields $R(f_{\theta^\star}) = R^\star = \inf_f R(f)$, and there is a constant $C_\phi$ depending only on $\phi$ such that if $|u_2| \leq C_\phi\beta|u_1|$, then*

$$\inf_t \mathbb{E}[\phi(Yt\langle u, X\rangle)] > \inf_\theta \mathbb{E}[\phi(Y\langle\theta, X\rangle)].$$

We turn to the proof of the lemma. Using the choices (26) of $x_1$ and $x_2$ and defining $s_1 = \langle u, x_1\rangle = \frac{1}{6}u_1$ and $s_2 = \langle u, x_2\rangle = \frac{u_1}{12\alpha}$, it follows that for any $t \in \mathbb{R}$,

$$\mathbb{E}_{P_{x_1,x_2}}[\phi(Y\langle tu, X\rangle)] = \frac{1}{2}\left(\frac{2}{3}\phi(ts_1) + \frac{1}{3}\phi(-ts_1)\right) + \frac{1}{2}\left(\frac{2}{3}\phi(ts_2) + \frac{1}{3}\phi(-ts_2)\right)$$

$$= \frac{1}{2}(g_\phi(ts_1) + g_\phi(ts_2)).$$

For $w_1, w_2 \in \mathbb{R}$, define the parameterized function

$$h_\phi(w_1, w_2) = \frac{1}{2}\inf_{t \in \mathbb{R}}\{g_\phi(tw_1) + g_\phi(tw_2)\},$$

which corresponds to the minimal value of the risk $t \mapsto \mathbb{E}_{P_{x_1,x_2}}[\phi(Y\langle t\theta, X\rangle)]$ when $w_1 = \langle\theta, x_1\rangle$ and $w_2 = \langle\theta, x_2\rangle$ for some vector $\theta \in \mathbb{R}^d$. The convexity and coercivity of $g_\phi$ imply that $h_\phi(w_1, w_2)$ is continuous on $\mathbb{R}^2 \setminus \{0\}$, it is homogeneous in that $h_\phi(tw_1, tw_2) = h_\phi(w_1, w_2)$ for all $t \neq 0$, and by construction,

$$\inf_{t \in \mathbb{R}} \mathbb{E}_{P_{x_1,x_2}}[\phi(Y\langle tu, X\rangle)] = h_\phi(s_1, s_2).$$

Moreover, it is immediate that

$$g_\phi^\star := \inf_t g_\phi(t) = \inf_{w_1^2 + w_2^2 = 1} \inf_t \frac{1}{2}\{g_\phi(tw_1) + g_\phi(tw_2)\} = \inf_{w_1^2 + w_2^2 = 1} h_\phi(w_1, w_2).$$

Let $\mathcal{G} = [a, b] = \mathrm{argmin}_t g_\phi(t)$, where we must have $0 < a \leq b < \infty$ as $\phi'(0) < 0$. Then we set the value $\alpha := \frac{b}{a} \geq 1$ in the definition (26) of the points $x_1, x_2$. Let $w_2 < \frac{1}{\alpha}w_1$; then if $w_1 \in \mathcal{G}$, we must have $w_2 < \frac{b}{\alpha} = a$, and so $w_2 \notin \mathcal{G}$, and so at least one of $w_1, w_2 \notin \mathcal{G}$. Enforcing the strict inequality $w_2 < \frac{1}{\alpha}w_1$, we see that

$$C_{\phi,\alpha} := \inf_{\substack{w_1^2 + w_2^2 = 1, \\ w_1 \geq 0, w_2 \leq \frac{3}{4\alpha}w_1}} h_\phi(w_1, w_2) > \inf_{w_1^2 + w_2^2 = 1} h_\phi(w_1, w_2) = g_\phi^\star.$$

Rewriting this in terms of the unit vector $u$ we have been considering, whenever $|u_2| \leq \frac{\beta u_1}{24\alpha(2\alpha - 1)}$,

$$s_2 \leq \frac{u_1}{12\alpha} + \frac{2\alpha - 1}{\beta} \cdot \frac{\beta u_1}{24\alpha(2\alpha - 1)} \leq \frac{u_1}{8\alpha} = \frac{3}{4\alpha}s_1,$$

and in this case

$$\inf_{t \in \mathbb{R}} \mathbb{E}_{P_{x_1,x_2}}[\phi(Y\langle tu, X\rangle)] = h_\phi(s_1, s_2) \geq \inf_{\substack{w_1^2 + w_2^2 = 1, \\ w_1 \geq 0, w_2 \leq \frac{3}{4\alpha}w_1}} h_\phi(w_1, w_2) = C_{\phi,\alpha} > g_\phi^\star.$$

Now we restrict $u \in \mathbb{S}^{d-1}$ to the collection of vectors satisfying $|u_2| \leq \frac{\beta}{24\alpha(2\alpha-1)}|u_1|$, and show that if $\theta \in \text{span}\{u\}$, the surrogate risk cannot attain its minimum. Indeed, recalling the construction (26), the matrix

$$\begin{bmatrix} 1/6 & -1/(12\alpha) \\ 0 & (2\alpha-1)/\beta \end{bmatrix}$$

is invertible and we can find a $\overline{\theta}$ such that $\langle \overline{\theta}, x_1 \rangle = c, \langle \overline{\theta}, x_2 \rangle = -c$ for some value $c \in \mathcal{G} = \arg\min g_\phi$, implying

$$\inf_{t \in \mathbb{R}} \mathbb{E}_P \left[ \phi(Y \langle tu, X \rangle) \right] \geq (1-\delta) \inf_{t \in \mathbb{R}} \mathbb{E}_{P_{x_1,x_2}} \left[ \phi(Y \langle tu, X \rangle) \right] = (1-\delta)C_{\phi,\alpha}$$

$$\inf_{\theta \in \mathbb{R}^d} \mathbb{E}_P \left[ \phi(Y \langle \theta, X \rangle) \right] \leq \mathbb{E}_P \left[ \phi(Y \langle \overline{\theta}, X \rangle) \right] = (1-\delta)g_\phi^\star + \delta \mathbb{E}_{\mathsf{N}(0,I_d)} \left[ \phi(Y \langle \overline{\theta}, X \rangle) \right].$$

By taking $\delta$ sufficiently small and using $C_{\phi,\alpha} > g_\phi^\star$, we conclude that Lemma F.1 holds with $C_\phi = \frac{1}{24\alpha(2\alpha-1)}$ and recognizing that $\alpha > \frac{1}{2}$ was otherwise arbitrary.

**Controlling the angle between $\theta_\varphi$ and $\theta^\star$.** By Lemma F.1, there exists a constant $C_\phi$ such that for any $\beta > 0$, we can construct a distribution $P$ for which any minimizer $\theta_\varphi$ of the surrogate risk must satisfy

$$|\langle \theta_\varphi, e_2 \rangle| \geq C_\phi \beta \cdot |\langle \theta_\varphi, e_1 \rangle| \quad \text{and} \quad |\langle \theta_\varphi, e_2 \rangle| > 0.$$

Now we specify the parameter $\beta$, taking $\beta = \frac{1}{C_\phi}\sqrt{\frac{1}{\epsilon^2} - 1}$. Then evidently

$$\langle \theta_\varphi, e_2 \rangle^2 \geq \left( \frac{1}{\epsilon^2} - 1 \right) \cdot \langle \theta_\varphi, e_1 \rangle^2,$$

which combined with $\theta^\star = e_1$ implies

$$|\cos \angle(\theta_\varphi, \theta^\star)| = \frac{|\langle \theta_\varphi, e_1 \rangle|}{\|\theta_\varphi\|_2} \leq \frac{|\langle \theta_\varphi, e_1 \rangle|}{\sqrt{|\langle \theta_\varphi, e_1 \rangle|^2 + |\langle \theta_\varphi, e_2 \rangle|^2}} \leq \frac{1}{\sqrt{1 + \frac{1}{\epsilon^2} - 1}} = \epsilon.$$

Because $\theta_\varphi \notin \text{span}\{\theta^\star\}$, we see that $R(f_{\theta_\varphi}) > R(f_{\theta^\star}) = \inf_f R(f)$, completing the proof of Proposition 5.

### F.2 PROOF OF THEOREM 3

Let $\mathcal{P}_k = \{p \in \mathbb{R}_+^k \mid \langle \mathbf{1}, p \rangle = 1\}$ by the probability simplex in $\mathbb{R}^k$. For $p \in \mathcal{P}_k$, define the risk gaps

$$\delta_\varphi(s,p) := \mathbb{E}_p[\varphi(s,Y)] - \inf_s \mathbb{E}_p[\varphi(s,Y)] \quad \text{and} \quad \delta_\ell(s,p) := \mathbb{E}_p[\ell(\mathsf{d}(s),Y)] - \inf_s \mathbb{E}_p[\ell(\mathsf{d}(s),Y)]$$

and the gap functional

$$\overline{\psi}_\varphi(\epsilon,p) := \inf_s \{\delta_\varphi(s,p) \mid \delta_\ell(s,p) \geq \epsilon\}.$$

By the assumption that $\varphi$ is consistent, it is immediate Steinwart (2007) that $\overline{\psi}_\varphi(\epsilon,p) > 0$ for all $p \in \mathcal{P}_k$ and $\epsilon > 0$. Moreover, consistency implies Zhang (2004b) that if $p_{(1)} \geq p_{(2)} \geq \cdots \geq p_{(k)}$ denotes the order statistics of $p \in \mathcal{P}_k$, when we define the subset

$$\mathcal{P}_{k,c} := \{p \in \mathcal{P}_k \mid p_{(1)} \geq p_{(2)} + c\}$$

of well-separated distributions, then for all $c > 0$ we have the strict inequality

$$\inf_{p \in \mathcal{P}_{k,c}} \overline{\psi}_\varphi(\epsilon,p) > 0 \quad \text{when } \epsilon > 0.$$

For $m \in \mathbb{N}$, let $P_m(\cdot \mid x)$ denote the induced distribution on the majority vote $Y_m^+ := \text{Majority}(Y_1^m)$ for $Y_i \overset{\text{iid}}{\sim} P(Y \in \cdot \mid X = x)$, so that if $\Delta(x) > 0$ we see that $Y_m^+ \to y^\star(x)$ with probability 1. Then

$$\delta_\ell(s, P_m(\cdot \mid x)) = \mathbb{E}_{P_m}[\mathbf{1}\{\mathsf{d}(s) \neq Y_m^+\} \mid x] - (1 - P(Y_m^+ = y^\star(x) \mid x))$$

$$= \begin{cases} 0 & \text{if } \mathsf{d}(s) = y^\star(x) \\ P(Y_m^+ = y^\star(x) \mid x) - P(Y_m^+ = \mathsf{d}(s) \mid x) & \text{otherwise.} \end{cases}$$

In particular, for $P$-almost-all $x$, we see that $\delta_\ell(s, P_m(\cdot \mid x)) \to 1\{\mathsf{d}(s) = y^\star(x)\}$ as $m \to \infty$. Now, fix $c > 0$ and define

$$\overline{\psi}_\varphi(\epsilon) := \inf_{p \in \mathcal{P}_{k,c}} \overline{\psi}_\varphi(\epsilon, p) \ \text{ and } \ \psi_\varphi(\epsilon) := \overline{\psi}_\varphi^{**}(\epsilon),$$

the convex conjugate of the gap functional on well-separated distributions. Then Zhang (2004b, Prop. 25) shows that $\overline{\psi}_\varphi^{**}(\epsilon) > 0$ whenever $\overline{\psi}_\varphi(\epsilon) > 0$.

We now consider the gaps in the surrogate risk $R_{\varphi,A_m}(f) - R_{\varphi,A_m}^\star$. Letting $c > 0$, define

$$\mathcal{X}_{c,m} := \left\{ x \mid P_m(Y_m^+ \in \cdot \mid X = x) \in \mathcal{P}_{k,c} \right\}$$

to be those $x \in \mathcal{X}$ for which the majority vote is likely correct. Then

$$R_{\varphi,A_m}(f) - R_{\varphi,A_m}^\star$$
$$= \mathbb{E}[\delta_\varphi(f(X), P_m(\cdot \mid X))]$$
$$\geq \mathbb{E}[\overline{\psi}_\varphi(\delta_\ell(f(X), P_m(\cdot \mid X)), P_m(\cdot \mid X))]$$
$$\geq \mathbb{E}\left[ 1\{X \in \mathcal{X}_{c,m}\} \psi_\varphi(\delta_\ell(f(X), P_m(\cdot \mid X))) + 1\{X \notin \mathcal{X}_{c,m}\} \overline{\psi}_\varphi(\delta_\ell(f(X), P_m(\cdot \mid X))) \right]$$
$$\geq \mathbb{E}\left[ 1\{X \in \mathcal{X}_{c,m}\} \psi_\varphi(\delta_\ell(f(X), P_m(\cdot \mid X))) \right].$$

Using Jensen's inequality that for any convex $h$, random variable $Z$, and set $A$, $\mathbb{E}[1\{Z \in A\} h(Z)] = \mathbb{E}[h(Z) \mid Z \in A]P(Z \in A) \geq h(\mathbb{E}[Z \mid Z \in A])P(Z \in A)$, we therefore obtain that

$$R_{\varphi,A_m}(f) - R_{\varphi,A_m}^\star \geq \psi_\varphi \left( \mathbb{E}\left[ \delta_\ell(f(X), P_m(\cdot \mid X)) \mid X \in \mathcal{X}_{c,m} \right] \right) P(X \in \mathcal{X}_{c,m})$$
$$= \psi_\varphi \left( \frac{R(f) - R^\star - \mathbb{E}[\delta_\ell(f(X), P_m(\cdot \mid X))1\{X \notin \mathcal{X}_{c,m}\}]}{P(X \in \mathcal{X}_{c,m})} \right) P(X \in \mathcal{X}_{c,m})$$
$$\geq \psi_\varphi \left( \frac{[R(f) - R^\star - P(X \notin \mathcal{X}_{c,m})]_+}{P(X \in \mathcal{X}_{c,m})} \right) P(X \in \mathcal{X}_{c,m}). \tag{27}$$

Let $R_{\varphi,\infty}(f) = \mathbb{E}[\varphi(f(X), y^\star(X))] = \lim_{m \to \infty} \mathbb{E}[\varphi(f(X), Y_m^+)]$. Then if $f^\star \in \mathcal{F}$ is any function with $\operatorname{argmax}_y f_y^\star(x) = y^\star(x)$ (for $P$-almost all $x$), we evidently obtain

$$\lim_{t \to \infty} R_{\varphi,\infty}(tf^\star) = 0$$

by dominated convergence, as by assumption (14) we have $\varphi(tf^\star(x), y^\star(x)) \to 0$ as $t \to \infty$ for almost all $x$. Let $\epsilon > 0$ be arbitrary and take any $t < \infty$ large enough that $R_{\varphi,\infty}(tf^\star) \leq \epsilon$. Then because $R_{\varphi,A_m}(tf^\star) \to R_{\varphi,\infty}$ as $m \to \infty$, for the sequence $f_m \in \epsilon_m$-$\operatorname{argmin} R_{\varphi,A_m}$, we obtain

$$R_{\varphi,A_m}(f_m) \leq R_{\varphi,m}(tf^\star) + \epsilon_m \to R_{\varphi,\infty}(tf^\star) \leq \epsilon.$$

Substituting into inequality (27), we have

$$\epsilon \geq \limsup_m R_{\varphi,A_m} - R_{\varphi,A_m}^\star \geq \limsup_m \psi_\varphi \left( \frac{[R(f_m) - R^\star - P(X \notin \mathcal{X}_{c,m})]_+}{P(X \notin \mathcal{X}_{c,m})} \right) P(X \in \mathcal{X}_{c,m}).$$

Because $P(X \notin \mathcal{X}_{c,m}) \to 0$ by assumption, if $\limsup_m R(f_m) - R^\star = \delta > 0$, we would obtain

$$\epsilon \geq \psi_\varphi(\delta).$$

But $\psi_\varphi(\delta) > 0$ for $\delta > 0$, and $\epsilon > 0$ was arbitrary, so it must be that $\limsup_m R(f_m) = R^\star$.

## G  PROOFS FOR MIS-SPECIFIED MODELS

### G.1  PROOF OF PROPOSITION 4

We assume the result of Theorem 2, as its proof does not depend on the current proposition. To simplify the proof and work with square matrices, we assume w.l.o.g. that $\Theta^\star = U^\star T^\star$, where $U^\star \in \mathbb{R}^{d \times (k-1)}$ is orthogonal, and we may w.l.o.g. take $T^\star$ to be diagonal, with $T^\star = \operatorname{diag}(t_1^\star, \ldots, t_{k-1}^\star)$, and let $\Theta_1(\epsilon) = U^\star T_1(\epsilon)$. It suffices to show that $T_1(\epsilon)/\|T_1(\epsilon)\| \neq T^\star/\|T^\star\|$. For simplicity, we suppress the dependence on $m = 1$ and write $T(\epsilon) = T_1(\epsilon)$, $\Theta(\epsilon) = \Theta_1(\epsilon)$, and let $T_{ij}(\epsilon)$ denote

the entries of $T(\epsilon)$. As $X \sim \mathsf{N}(0, I_d)$, it follows that $U^{\star\top}X \sim \mathsf{N}(0, I_{k-1})$, so that the stationary conditions for $\Theta(\epsilon)$ equivalently state that for $Z \sim \mathsf{N}(0, I_{k-1})$,

$$\nabla_\Theta L_{1,\epsilon}(\Theta(\epsilon)) = \mathbb{E}\left[Z\left(\sigma^{\mathrm{lr}}(T(\epsilon)^\top Z) - \sigma^\epsilon(T^{\star\top}Z)\right)^\top\right] = \mathbf{0}_{d\times(k-1)}.$$

Let $\mathcal{T} \subset \mathbb{R}^{k-1}$ be a set to be chosen, and write $\sigma(t) = \sigma^{\mathrm{lr}}(t)\mathbf{1}\{t \notin \mathcal{T}\} + \sigma^{\mathrm{uni}} \cdot \mathbf{1}\{t \in \mathcal{T}\}$, where $\sigma^{\mathrm{uni}} = \frac{1}{k}\mathbf{1}$ denotes the uniform distribution. Then equivalently,

$$\mathbb{E}\left[Z\left(\sigma^{\mathrm{lr}}(T(\epsilon)^\top Z) - \sigma^{\mathrm{lr}}(T^{\star\top}Z)\right)^\top\right] + \underbrace{\mathbb{E}\left[Z\left(\sigma^{\mathrm{lr}}(T^{\star\top}Z) - \sigma^{\mathrm{uni}}\right)^\top \mathbf{1}\left\{T^{\star\top}Z \in \mathcal{T}\right\}\right]}_{=:A(\mathcal{T})} = \mathbf{0}.$$

For small $\epsilon > 0$, we can always choose disjoint $\mathcal{T}_\epsilon$ and $\mathcal{T}_{-\epsilon}$ with $P(T^{\star\top}Z \in \mathcal{T}_\epsilon), P(T^{\star\top}Z \in \mathcal{T}_{-\epsilon}) \leq \epsilon$ while the matrices $A(\mathcal{T}_\epsilon)$ and $A(\mathcal{T}_{-\epsilon})$ belong to distinct rays, that is, are not positive multiples of one another. Indeed, as the rank one matrix $T^{\star\top}Z(\sigma^{\mathrm{lr}}(T^{\star\top}Z) - \sigma^{\mathrm{uni}})^\top$ is non-constant whenever $k \geq 3$, we can find a matrix $Q \in \mathbb{R}^{(k-1)\times(k-1)}$ such that the sets

$$\mathcal{T}_+ = \left\{T^{\star\top}z \mid \mathrm{tr}\left(Q^\top T^{\star\top}z(\sigma^{\mathrm{lr}}(T^{\star\top}z) - \sigma^{\mathrm{uni}})^\top\right) > 0\right\}$$

$$\mathcal{T}_- = \left\{T^{\star\top}z \mid \mathrm{tr}\left(Q^\top T^{\star\top}z(\sigma^{\mathrm{lr}}(T^{\star\top}z) - \sigma^{\mathrm{uni}})^\top\right) < 0\right\}$$

have positive Lebesgue measure. Then for any $\mathcal{T}_\epsilon \subset \mathcal{T}_+$ and $\mathcal{T}_{-\epsilon} \subset \mathcal{T}_-$, we must have $\mathrm{tr}(Q^\top A(\mathcal{T}_\epsilon)) > 0$ and $\mathrm{tr}(Q^\top A(\mathcal{T}_{-\epsilon})) < 0$, as desired, and we may take the sets $\mathcal{T}_{\pm\epsilon}$ to have Lebesgue measure at most $\epsilon$. By absolute continuity of Lebesgue integral, as $\epsilon \to 0$ it follows $A(\mathcal{T}_\epsilon) \to 0$ and $A(\mathcal{T}_{-\epsilon}) \to 0$ uniformly with $\epsilon$.

Now we are ready to prove the lemma. Consider the tilted gradient function

$$F(T, A) = \mathbb{E}\left[Z\left(\sigma^{\mathrm{lr}}(T^\top Z) - \sigma^{\mathrm{lr}}(T^{\star\top}Z)\right)^\top\right] + A,$$

which satisfies $F(T^\star, 0) = 0$, and for which the linear mapping

$$D(T) = \nabla_T F(T, 0) : \mathbb{R}^{(k-1)\times(k-1)} \to \mathbb{R}^{(k-1)\times(k-1)}, \quad D(T)[M] := \mathbb{E}[Z(\nabla\sigma^{\mathrm{lr}}(T^\top Z)MZ)^\top]$$

is invertible at $D(T^\star)$. By construction of the matrices $A_{\pm\epsilon} := A(\mathcal{T}_{\pm\epsilon})$, we also know that there exist solutions $T(\pm\epsilon)$ satisfying $F(T(\epsilon), A_\epsilon) = 0$ and $F(T(-\epsilon), A_{-\epsilon}) = 0$. By the implicit function theorem and that $\nabla_A F(T, A) = \mathrm{Id}$, we thus obtain

$$T(\epsilon) = T^\star - D(T^\star)^{-1}\nabla_A F(T^\star, 0)A_\epsilon + O(\|A_\epsilon\|^2)$$

$$= T^\star - D(T^\star)^{-1}A_\epsilon + O(\|A_\epsilon\|^2),$$

and similarly $T(-\epsilon) = T^\star - D(T^\star)^{-1}A_{-\epsilon} + O(\|A_{-\epsilon}\|^2)$. Without explicitly computing the Jacobian, we may still conclude that at least one of $T(\epsilon)$ and $T(-\epsilon)$ cannot align with $T^\star$, as $T(\epsilon) - T^\star$ and $\overline{T}(\epsilon) - T^\star$ belong to distinct rays.

### G.2 PROOF OF THEOREM 2

We prove the theorem in two parts. In the first we verify the validity of the ansatz $\Theta_m = U^\star T_m$, and in the second we show the claimed asymptotics of $T_m$.

**Part 1: Ansatz for the population loss.** Let $Z = U^{\star\top}X \sim \mathsf{N}(0, U^{\star\top}\Sigma U^\star)$, and let $A \in \mathbb{R}^{d\times(k-1)}$ satisfy

$$0 = \mathrm{Cov}(X - AZ, Z) = \Sigma U^\star - AU^{\star\top}\Sigma U^\star,$$

i.e., $A = \Sigma U^\star(U^{\star\top}\Sigma U^\star)^{-1}$. Then $X - AZ$ and $Z$ are independent. Consider the lower dimensional problem in $\mathbb{R}^{k-1}$ with the covariates $X$ replaced by $Z$ and $\Theta^\star$ replaced by $T^\star$, with associated loss (abusing notation)

$$\overline{L}_m(T) := \mathbb{E}\left[-\log P_T(Y_m^+ \mid Z)\right] = \mathbb{E}\left[\varphi(T^\top Z, Y_m^+)\right],$$

where $Y_m^+$ denotes majority vote and $P_T$ the logistic regression model. The loss $L_m$ is still strictly convex and coercive, so it has unique minimum $T_m \in \mathbb{R}^{(k-1)\times(k-1)}$ satisfying

$$\nabla_\Theta \overline{L}_m(T_m) = \mathbb{E}\left[Z(\sigma^{\mathrm{lr}}(T_m^\top Z) - \rho_m(T^{\star\top} Z))^\top\right] = 0,$$

where we recall the notation that $\rho_m(v) = (P(Y_m^+ = 1), \ldots, P(Y_m^+ = k))$ when $Y_i \overset{\mathrm{iid}}{\sim} \mathsf{Cat}(v)$.

We demonstrate $\Theta_m = U^\star T_m$ minimizes $L_m$. Indeed,

$$\nabla_\Theta L_m(\Theta_m) = \mathbb{E}\left[X\left(\sigma^{\mathrm{lr}}(T_m^\top Z) - \rho_m(T^{\star\top} Z)\right)^\top\right]$$

$$= A\, \underbrace{\mathbb{E}\left[Z\left(\sigma^{\mathrm{lr}}(T_m^\top Z) - \rho_m(T^{\star\top} Z)\right)^\top\right]}_{=0 \text{ by stationarity of } \overline{L}_m} + \mathbb{E}\left[(X - AZ)\left(\sigma^{\mathrm{lr}}(T_m^\top Z) - \rho_m(T^{\star\top} Z)\right)^\top\right]$$

$$\overset{(\star)}{=} \underbrace{\mathbb{E}[X - AZ]}_{=0} \cdot \mathbb{E}[\sigma^{\mathrm{lr}}(T_m^\top Z) - \rho_m(T^{\star\top} Z)]^\top = 0,$$

where equality $(\star)$ uses the independence between $X - AZ$ and $Z$.

**Part 2: Asymptotics of $T_m$.**  We prove $\|T_m\| \to \infty$ and $T_m/\|T_m\| - T^\star/\|T^\star\| \to 0$.

**Lemma G.1.** *Under the conditions of the theorem, $\Theta_m = \arg\min_\Theta L_m(\Theta)$ satisfies $\|\Theta_m\|_{\mathrm{op}} = \|T_m\|_{\mathrm{op}} \to \infty$ and $L_m(\Theta_m) \to 0$.*

**Proof**  When $\|T\|_{\mathrm{op}} \le r$, for $\Theta = U^\star T$ and $i, j \in [k]$ we have

$$|\langle \theta_i - \theta_j, x\rangle| = |\langle U^\star T(e_i - e_j), x\rangle| \le \|e_i - e_j\|_2 \|T\|_{\mathrm{op}} \|x\|_2 \le \sqrt{2}r \|x\|_2$$

Therefore we have pointwise lower bound for the loss

$$\varphi(\Theta^\top x, y) = \log\left(\sum_{i=1}^k \exp(\langle \theta_i - \theta_y, x\rangle)\right) \ge \log\left(1 + (k-1)\exp\{-\sqrt{2}r\|x\|_2\}\right) > 0.$$

Letting $g(r) := \mathbb{E}[\log(1 + (k-1)\exp(-\sqrt{2}r\|X\|_2))] > 0$, which is a strictly decreasing function of $r$, we see that for all $m \in \mathbb{N}$ and $\|\Theta\|_{\mathrm{op}} = \|T\|_{\mathrm{op}} \le r$, $L_m(\Theta) \ge g(r)$.

On the other hand, for a real number $R > 0$, consider $\Theta_R := R\Theta^\star/\|\Theta^\star\|_{\mathrm{op}}$, whose columns $\theta_1, \ldots, \theta_k$ are scaled multiples of those of $\Theta^\star$. It is clear from majority vote consistency that $\rho_m(\Theta_R x) \to e_{y^\star(x)}$ as $m$ or $R \to \infty$, and so

$$L_m(\Theta_R) = \mathbb{E}[\varphi(\Theta_R^\top X, Y_m^+)]$$

$$\underset{m\uparrow\infty}{\longrightarrow} \mathbb{E}\left[\log\left(\sum_{i=1}^k \exp(\langle \theta_i - \theta_{y^\star(X)}, X\rangle)\right)\right]$$

$$\le \mathbb{E}\left[\log\left(1 + (k-1)\exp\left(-R\min_{i\ne j}\|\theta_i^\star - \theta_j^\star\|_2 \|X\|_2 / \|T^\star\|_{\mathrm{op}}\right)\right)\right] =: h(R).$$

We conclude that

$$\limsup_m \inf_\Theta L_m(\Theta) \le h(R)$$

This implies for sufficiently large $m$, $\inf_\Theta L_m(\Theta) < 2h(R)$ and we must have $\|\Theta_m\| \ge g^{-1}(2h(R))$. As both $g$ and $h$ monotonically decrease to 0 on $\mathbb{R}_+$, we see that $\|\Theta_m\| \to \infty$. The unitary invariance of $\|\cdot\|_{\mathrm{op}}$ gives that $\|\Theta_m\|_{\mathrm{op}} = \|T_m\|_{\mathrm{op}}$, and that $h(R) \to 0$ as $R \uparrow \infty$ implies $L_m(\Theta_m) \to 0$.  $\square$

We now demonstrate the asymptotic alignment $T_m/\|T_m\| - T^\star/\|T^\star\| \to 0$. Define the mis-aligned region

$$\mathcal{R}(\epsilon) := \{T \mid \|T/\|T\| - T^\star/\|T^\star\|\| \ge \epsilon\}.$$

Let $\Theta = U^\star T$ for some $T \in \mathcal{R}(\epsilon)$, and define the set

$$\mathcal{X}(T) := \left\{ x \in \mathcal{X} \mid \operatorname*{argmax}_y \langle \theta_y, x \rangle \neq \operatorname*{argmax}_y \langle \theta_y^\star, x \rangle \right\}.$$

Then we have the lower bound

$$L_m(\Theta) = \mathbb{E}[\varphi(\Theta^\top X, Y_m^+)] \geq \mathbb{E}[\varphi(\Theta^\top X, Y_m^+)1\{X \in \mathcal{X}(T)\}]$$

$$\geq \mathbb{E}\left[ e_{y^\star(X)}^\top \rho_m(\Theta^{\star\top} X) \log \left( 1 + \sum_{j \neq y^\star(X)} \exp(\langle \theta_j - \theta_{y^\star(X)}, X \rangle) \right) 1\{X \in \mathcal{X}(T)\} \right]$$

$$\geq \log 2 \cdot \mathbb{E}\left[ e_{y^\star(X)}^\top \rho_m(\Theta^{\star\top} X) 1\{X \in \mathcal{X}(T)\} \right],$$

where we use that on the set $x \in \mathcal{X}(T)$, at least one column $\theta_j$ satisfies $\langle \theta_j - \theta_{y^\star(x)}, x \rangle \geq 0$. By dominated convergence, as $m \to \infty$,

$$\liminf_m L_m(\Theta) \geq \log 2 \cdot P(X \in \mathcal{X}(T)).$$

Because $\mathcal{X}(T)$ is a union of subspaces, $T \mapsto P(X \in \mathcal{X}(T))$ is continuous and homogeneous in $\|T\|$, so that $\inf_{T \in \mathcal{R}(\epsilon)} P(X \in \mathcal{X}(T)) > 0$.

We have thus shown that $\liminf_m \inf_{\Theta \in U^\star \mathcal{R}(\epsilon)} L_m(\Theta) > 0$. However, Lemma G.1 shows that $\|\Theta_m\| \to \infty$ and $L_m(\Theta_m) \to 0$, so we must have $\Theta_m \notin U^\star \mathcal{R}(\epsilon)$ for large $m$, and so $T_m / \|T_m\| \to T^\star / \|T^\star\|$.

