# OpenReview forum: "Some Robustness Properties of Label Cleaning"
_ICLR.cc/2026/Conference — ICLR 2026 Conference Withdrawn Submission_

### Official Review · Reviewer_xdn5 · 2025-10-31

**Soundness:** 1
**Presentation:** 1
**Contribution:** 3
**Rating:** 4
**Confidence:** 2

**Summary:**

The work takes a deeper analysis on the robustness of surrogate losses using label aggregation like majority voting. The paper argues that aggregating multiple noisy labels before training yields robustness and consistency guarantees that we cannot achieve when learning directly from single raw labels. It shows several simple examples from ranking to multi-class classification to support the theoretical results presented. The paper, rich in analysis, does not present any empirical evidence connecting to the practicality of the observations. Some simple demonstrations could have been added to support the theoretical results.

**Strengths:**

Strengths:

1.	The paper takes a deep statistical learning theoretic approach to the label denoising task, substantiating many empirical successes of the label aggregation algorithms such as majority voting.

2.	The analysis such as used in ranking could be beneficial to drive more in-depth look in how this may be useful in current popular frameworks such as RLHF.

**Weaknesses:**

See Questions section.

**Questions:**

1.	The organization of the paper needs quite a bit of improvement. The logical flow is missing in several places where discussion is jumping from one example to another, like starting with ranking to then switching classification etc without providing the reasoning of the shift.

2.	The paper attempts to investigate “robustness” of label cleaning. Robustness can have many interpretations and definitions in learning. There is alack of clarity in what sort of robustness the authors are investigating. Is it task-specific or model-specific?

3.	The introduction section talks about comparing label cleaning and aggregation, but in my opinion, the discussions are bit opaque in describing the setup whether the context is with using crowdsourced labels/multiple labels.

4.	After Corrollary 3, the discussion is around whether using paired observations (X,Y) or tuples with multiple labels for Fisher consistency. However, the analysis in section 3 does not reveal any such comparison. It is unclear why “using only paired observations (X, Y ) rather than tuples (X, Y1, . . . , Ym), we could bring the entire theory of empirical processes and related statistical tools”. For example, the works in [1] has presented some analysis using such noisy tuples establishing statistical consistency and error bounds. So, it is unclear what are specific analytical challenges here?

[1] Ibrahim, Shahana, Tri Nguyen, and Xiao Fu. "Deep learning from crowdsourced labels: Coupled cross-entropy minimization, identifiability, and regularization, ICLR 2023.

5.	It is not clear what is “*” after Proposition 1 in Section 3.1. What is the implication of Proposition 1 and 2? Why Proposition 2 considers only a certain definition of the surrogate loss? Why this observation (fisher consistency failure without label aggregation) happens for ranking, but not in classification? Alos, this results shows convex surrogate losses, so how does this result impact/inform the practically used surrogate losses, which need not be convex?

6.	Example 3 looks trivial, as the Definition 3.1 is based on label aggregation. Does the example cover the multiple label case as well?

7.	The asymptotic analysis with m going to infinity is a bit impractical assumption. Most asymptotic analysis in the existing studies looks at number of observations is going to be infinite. I see the discussion in the last part of the paper acknowledges this to some extent. But there is lack of clarity in the nature of asymptotic analysis used here. Does that mean repeatedly sampling infinitely for observations or unique labelers (as in crowdsourcing scenario)?

8.	In Eq. (8), y^* is not defined.

9.	In section 4.1, optimal linear predictors are analyzed. How does the label noise rate in fact affect this analysis? The definition of optimality is a bit unclear in this section? Does this change with level of label noise?

---

### Official Review · Reviewer_SAh9 · 2025-11-01

**Soundness:** 2
**Presentation:** 2
**Contribution:** 2
**Rating:** 2
**Confidence:** 2

**Summary:**

This paper investigates the theoretical “robustness” properties of label aggregation, arguing that combining multiple noisy labels via averaging or majority voting can enhance surrogate risk consistency and yield more robust classifiers. The authors formalize this through a series of asymptotic results involving Fisher consistency, identifying surrogates, and consistency.

While the general theme of robust learning under noisy supervision is relevant, the paper contributes little new theoretical insight. The results largely restate well-known intuitions that aggregating noisy labels reduces variance and improves consistency without offering novel analysis, quantitative bounds, or practical implications. The exposition is heavy and unfocused, and the theory remains purely asymptotic and disconnected from realistic labeling scenarios. Overall, the work is technically shallow, poorly motivated, and lacks both novelty and practical significance. I recommend rejection.

**Strengths:**

- Addresses a broadly relevant theme: robustness under label noise.
- Connects to known frameworks on surrogate risk consistency (e.g., Bartlett & Jordan, Steinwart).
- Includes some formal statements and asymptotic guarantees.

**Weaknesses:**

- The idea that aggregating noisy labels (where noise rate is not dominant) improves robustness is self-evident and already well known. Most theorems are restatements of standard asymptotic arguments; if you average away noise, you approach the Bayes optimal predictor.
- Lack of practical or scientific significance. Most results assume the number of labels per example $m\rightarrow \infty$, which is unrealistic. No finite-m or sample-complexity analysis is provided. The setting is detached from any real-world labeling scenario.
- Poor exposition and organization. The writing is unclear, repetitive, and full of unnecessary formalism. The narrative is disjointed, making it difficult to follow the motivation or intuition. It often reads as though auto-generated or assembled from prior papers rather than written coherently.
- No empirical or illustrative evidence. Despite focusing on robustness, the paper provides no empirical demonstrations or even synthetic experiments to illustrate its claims. The absence of finite-sample evidence severely undermines the credibility and practical impact of the theoretical results.

**Questions:**

- Can the authors provide any finite-m characterization of when aggregation actually helps?
- Is there any empirical evidence that the derived asymptotic trends manifest in realistic datasets?

---

### Official Review · Reviewer_x3t4 · 2025-11-01

**Soundness:** 3
**Presentation:** 3
**Contribution:** 3
**Rating:** 6
**Confidence:** 3

**Summary:**

This paper studies the theoretical benefits of label aggregation (e.g., majority voting over multiple noisy labels) for supervised learning. The authors argue that data cleaning—far from being merely a preprocessing step—fundamentally enhances surrogate consistency and robustness in ways that are impossible with single noisy labels.

**Strengths:**

- The work shows situations where no convex surrogate is Fisher-consistent without aggregation. These results are compelling and motivate aggregation theory.
- The logistic-regression example is striking: m-majority vote converges to the true direction with mis-specified link, while raw labels fail—even under tiny label corruption.
- The paper contextualizes within seminal work (Bartlett et al., Steinwart, Tsybakov), providing a credible extension.

**Weaknesses:**

- Given the theoretical nature this is understandable, but even small-scale simulations (ranking / multiclass) would aid intuition.
- Majority vote is treated as the canonical aggregator. Extending beyond simple voting would broaden impact.

**Questions:**

The conditions (e.g., identifying surrogate, noise functions, κ(x)) are mathematically elegant but not intuitive for practitioners.

---

### Official Review · Reviewer_nKCk · 2025-11-02

**Soundness:** 3
**Presentation:** 3
**Contribution:** 3
**Rating:** 4
**Confidence:** 2

**Summary:**

This paper studies the robustness of learning with aggregated labels (e.g., majority vote) and establishes that label aggregation provides stronger surrogate risk consistency than using raw noisy labels. The authors show that standard Fisher consistency can fail in tasks like ranking or multiclass classification, but aggregation can “upgrade” inconsistent surrogates into consistent ones. They prove new theoretical results demonstrating consistency amplification and robustness, even for finite-dimensional or slightly mis-specified models. Overall, the work reframes label cleaning as a theoretically grounded robustness mechanism rather than a heuristic preprocessing step, supported by rigorous analysis and general theorems.

**Strengths:**

The paper offers rigorous extensions of surrogate risk consistency theory, introducing new definitions (e.g., identifying surrogates) and proving nontrivial consistency amplification results.

By interpreting label cleaning as a theoretical mechanism for robustness, the paper bridges practical data processing (e.g., crowdsourcing denoising) and formal consistency theory.

The framing of label aggregation as robustness enhancement, rather than a heuristic data cleaning step, is compelling and original.

**Weaknesses:**

The paper is entirely theoretical; it would benefit from even minimal empirical demonstrations or synthetic experiments showing how label aggregation improves performance in practice.

The mathematical exposition is dense and heavily notation-driven. The core intuition, why aggregation improves consistency, could be explained more visually or intuitively.

The theory focuses primarily on convex surrogates and majority-vote-like aggregations; it is unclear how well results generalize to modern deep-learning setups or non-convex training objectives.

The connection to real-world label noise processes (e.g., annotator bias, adversarial noise) could be more concretely illustrated.

**Questions:**

Can the authors provide empirical verification (e.g., on crowdsourced or noisy benchmark datasets) to support their theoretical robustness claims?

Are there settings where aggregation could harm performance—for example, when label noise is adversarial or when aggregation over-smooths minority-label signals?

---

### Note · Authors · 2025-11-18

**Comment:**

Dear AC(s) and Reviewers:

We have decided to withdraw this paper, as it appears that ICLR is not the correct venue for it. We appreciate that the reviewers have taken the time to review the paper. A few points follow to explain our decision:

(i) The reviewers all have low confidence scores and further revision seems pointless for both the authors and reviewers.

(ii) Our paper is purely theoretical, with dense results. Based on the reviewer feedback, it does not seem the ICLR audience will appreciate this.

(iii) The review by SAh9 is, frankly, somewhat offensive, and certainly lacks evidence for their assertions. For example, the claim that “Most theorems are restatements of standard asymptotic arguments” is incorrect, as Theorems 1 and 2 are both new (unless the reviewer has evidence that they are not; such a claim certainly requires justification). Similarly, that “aggregating noisy labels […] improves robustness is self-evident and already well known” is the kind of statement that, at least in our writing, would require citation. For example, the papers https://arxiv.org/abs/2402.02249 and https://arxiv.org/abs/2206.12041 both seem to argue that this is not so self-evident. Perhaps we have misunderstood what is considered an effective and accurate review with appropriate supporting evidence. The statement that the “paper reads as though auto-generated” is insulting.

Authors

**Withdrawal Confirmation:**

I have read and agree with the venue's withdrawal policy on behalf of myself and my co-authors.